# ComRank: Ranking Loss for Multi-Label Complementary Label Learning

**Jing-Yi Zhu**[1,2], **Yi Gao**[1,2*], **Miao Xu**[3], **Min-Ling Zhang**[1,2]
[1]School of Computer Science and Engineering, Southeast University, Nanjing 210096, China
[2]Key Laboratory of Computer Network and Information Integration (Southeast University),
Ministry of Education, China
[3]The University of Queensland, Australia
{zhujingyi, gao_yi, zhangml}@seu.edu.cn,
{miao.xu}@uq.edu.au

## Abstract

Multi-label complementary label learning (MLCLL) is a weakly supervised paradigm that addresses multi-label learning (MLL) tasks using complementary labels (i.e., irrelevant labels) instead of relevant labels. Existing methods typically adopt an unbiased risk estimator (URE) under the assumption that complementary labels follow a uniform distribution. However, this assumption fails in real-world scenarios due to instance-specific annotation biases, making URE-based methods ineffective under such conditions. Furthermore, existing methods underutilize label correlations inherent in MLL. To address these limitations, we propose **ComRank**, a ranking loss framework for MLCLL, which encourages complementary labels to be ranked lower than non-complementary ones, thereby modeling pairwise label relationships. Theoretically, our surrogate loss ensures Bayes consistency under both uniform and biased cases. Experiments demonstrate the effectiveness of our method in MLCLL tasks. The code is available at https://github.com/JellyJamZhu/ComRank.

## 1 Introduction

Multi-label learning (MLL) refers to a task where an instance is associated with multiple relevant labels, which has broad applications in real-world scenarios [Zhang and Zhou, 2014, Tang et al., 2023, Kou et al., 2024]. However, accurately labeling a large number of instances with all their true labels incurs high labor costs. To address this issue, weakly supervised learning for MLL has gained widespread attention in recent years, including *semi-supervised multi-label learning* [Liu et al., 2006, Niu et al., 2019], *multi-label learning with missing labels* [Sun et al., 2010, Wu et al., 2014], and *partial multi-label learning* (PML) [Xie and Huang, 2018, Zhang and Fang, 2020].

Multi-label complementary label learning (MLCLL) has recently emerged as a weakly supervised learning paradigm, which enables algorithms to learn from complementary labels instead of relevant labels to address the MLL problem. In MLCLL, each training instance is associated with a complementary label rather than relevant labels, where the complementary label specifies a label that the instance **does not** belong to. One application for MLCLL is privacy preservation, such as in sensitive surveys where respondents may hesitate to provide all truthful answers. By only asking them to exclude certain options, data collection becomes easier and more privacy-friendly, while also reducing labeling costs.

---

*Corresponding author

39th Conference on Neural Information Processing Systems (NeurIPS 2025).

The conventional solution for MLCLL currently revolves around the unbiased risk estimator (URE), which is a powerful tool in weakly supervised learning that enables the accurate estimation of true classification risk. Specifically, these methods derive URE by assuming that complementary labels follow uniform distribution, where a URE can be constructed based on common MLL loss functions such as binary cross-entropy loss (BCE), mean squared error loss (MSE), and mean absolute error loss (MAE). With the above uniform assumption, Gao et al. [2023] firstly investigate the MLCLL problem and derive a URE. Moreover, they propose a gradient-friendly MLCLL loss to enhance gradient updating of the URE. However, this uniform assumption may fail in real-world scenarios, where annotators may provide complementary labels with biases influenced by the characteristics of the instances. Moreover, previous URE-based methods do not fully exploit label correlations, which are critical in MLL.

In MLCLL, based on the fact that complementary labels are known to be irrelevant, while non-complementary labels may include relevant ones, it is generally desirable for the predicted probabilities of complementary labels to rank lower than those of non-complementary labels. This intuition naturally aligns with the objective of ranking loss. Inspired by this observation, we propose a **complementary ranking loss** (called **ComRank**) framework for MLCLL, which encourages learning by enforcing this ranking constraint and capturing pairwise relationships between labels. Additionally, our framework uses an exponential loss as the surrogate loss, while the complementary ranking loss achieves Bayes consistency under both cases of uniform and biased complementary labels. This overcomes the limitation of the URE, which only has theoretical guarantees under uniform complementary labels. Furthermore, our proposed framework can directly capture label correlation information from the rankings, offering unique advantages in MLL. Outstanding experimental results demonstrate the effectiveness of our method. The main contributions of this paper are as follows:

- We theoretically analyze why existing URE-based methods cannot work well on complementary labels with a biased distribution. URE strongly depends on the uniform assumption of complementary labels, and fails to estimate the expected risk when the complementary label distribution shifts.

- We firstly introduce ranking loss into MLCLL and propose a complementary ranking loss framework, ComRank, for MLCLL. We theoretically prove that it possesses Bayes consistency under both uniform and biased complementary labels. Experiments on different complementary label distributions demonstrate the outstanding performance of our method.

The remainder of this paper is organized as follows: Section 2 summarizes the related work, and preliminaries are introduced in section 3. Discussion on URE with different complementary label distribution are provided in section 4. Section 5 and section 6 introduce ComRank and its experimental results. The last section 7 concludes our work.

## 2 Related Work

**Multi-label learning.** MLL is a classification task where each instance can be related to multiple labels simultaneously [Jia et al., 2023, Shi et al., 2024]. Considering label correlations during training, there has been extensive theoretical exploration of ranking in MLL. Gao and Zhou [2011] first defined the consistency of MLL and Li et al. [2017] introduced the concept of Bayes consistency into the context of ranking loss for MLL, proposing a surrogate loss proven to be consistent from the perspective of the Bayes prediction rule. Xie and Huang [2018] shows that in probabilistic MLL, ranking between positive and negative labels can help disambiguate false positives. Li et al. [2024] illustrates that missing labels can be assumed to rank between positive and negative labels in weakly supervised MLL settings. However, these studies on ranking loss are all focused on supervised scenarios and are not applicable to the MLCLL problem.

**Complementary label learning (CLL).** CLL was first proposed to solve the multi-class classification tasks, which aims to train a multi-class classifier using complementary labels. Ishida et al. [2017] initially derived a URE by modifying pairwise-comparison and one-versus-all losses for CLL under the assumption of uniform complementary label distribution. To get rid of trapping in specific losses, Ishida et al. [2019] extended a general URE framework that can accommodate arbitrary losses. Recognizing that the uniform assumption for generating complementary labels may fail to handle the real-world scenarios, various methods that diverge from uniform assumption have also been

investigated, such as Yu et al. [2018] relaxed CLL to biased complementary labels by estimating a transition matrix, and Gao and Zhang [2021] developed a discriminative model without the uniform assumption. The success of these methods is based on the multi-class scenarios, but they may not be suitable for the MLCLL problem.

**Multi-label complementary label learning.** MLCLL was first introduced by Gao et al. [2023] as a solution to the challenge of collecting and accurately annotating multi-label data. Under the uniform assumption (i.e., randomly sampled from irrelevant labels), Gao et al. [2023] derived a URE, which ensures the classifier learned through complementary labels converges to the optimal classifier in MLL. Furthermore, in later research, Gao et al. [2024] allowed the use of biased complementary labels to recover relevant labels through an estimated transition matrix, and Gao et al. [2025] further investigated the URE in the setting of multiple complementary labels. Unfortunately, these methods fail to consider the biased complementary label distribution, or do not provide theoretical guarantees to derive a URE under biased distribution.

Therefore, in this paper, we demonstrate why URE-based methods cannot remain unbiased across non-uniform complementary label distributions, making it impossible for these methods to accurately estimate the true classification risk in MLL. Furthermore, we propose a novel complementary ranking loss framework, ComRank, for MLCLL, which offers theoretical guarantees under both uniform and biased complementary label distributions.

## 3 Preliminaries

In MLL, $\mathcal{X} \subset \mathbb{R}^d$ and $\mathcal{Y} \subseteq \{0,1\}^K$ represent the $d$-dimensional feature space and the label space with $K$ labels, respectively. A multi-label sample can be represented as $(\boldsymbol{x}, Y) \in \mathcal{X} \times \mathcal{Y}$, where $\boldsymbol{x}$ and $Y$ follow the probability distribution $p(\boldsymbol{x}, Y)$. Let the training set be $D = \{(\boldsymbol{x}_i, Y_i) \,|\, 1 \leq i \leq N\}$. Here, $Y$ can be written as a $K$-dimensional vector $\boldsymbol{y} = [y^1, y^2, \ldots, y^K] \in \{0,1\}^K$, where $y^k = 1$ indicates that the label $k$ is related to $\boldsymbol{x}$. The objective of MLL is to train a multi-label classifier $\boldsymbol{g} : \mathcal{X} \to \mathbb{R}^K$ by minimizing the following expected risk $R(\boldsymbol{g})$, where $g_k \in [0,1]$ is the predicted probability for the $k$-th element and $\mathcal{L}$ denotes the common MLL loss:

$$\boldsymbol{g}^* = \arg\min_{\boldsymbol{g} \in G} R(\boldsymbol{g}) = \arg\min_{\boldsymbol{g} \in G} \mathbb{E}_{p(\boldsymbol{x}, Y)} \left[ \mathcal{L}(\boldsymbol{g}(\boldsymbol{x}), Y) \right]. \tag{1}$$

In MLCLL, we define the complementary training set as $\bar{D} = \{(\boldsymbol{x}_i, \bar{y}_i) \,|\, 1 \leq i \leq N\}$, where $\bar{y}_i \in \{\mathcal{Y} - Y_i\}$. To ensure the validity of $\bar{y}_i$, $Y_i$ cannot be $\emptyset$ or $\mathcal{Y}$, so $|\mathcal{Y}| = 2^K - 2$. $\boldsymbol{x}$ and $\bar{y}$ follow a distribution $p(\boldsymbol{x}, \bar{y})$. For convenience, $\bar{y}$ can be represented as a $K$-dimensional vector $\bar{\boldsymbol{y}} = [\bar{y}^1, \bar{y}^2, \ldots, \bar{y}^K] \in \{0,1\}^K$, where $\bar{y}^k = 1$ indicates that the label $k$ is the complementary label of $\boldsymbol{x}$. MLCLL aims to train a multi-label classification classifier $\bar{\boldsymbol{g}} : \mathcal{X} \to \mathbb{R}^K$, which can predict relevant labels for unseen instances. Let $\bar{\mathcal{L}}$ represent the MLCLL loss, and the optimal classifier $\bar{\boldsymbol{g}}^*$ is obtained by minimizing the expected risk of MLCLL:

$$\bar{\boldsymbol{g}}^* = \arg\min_{\boldsymbol{g} \in G} \bar{R}(\boldsymbol{g}) = \arg\min_{\boldsymbol{g} \in G} \mathbb{E}_{p(\boldsymbol{x}, \bar{y})} \left[ \bar{\mathcal{L}}(\boldsymbol{g}(\boldsymbol{x}), \bar{y}) \right]. \tag{2}$$

Before commencing the analysis, we first present the definitions of URE. URE is an important tool for weakly supervised method [Ishida et al., 2019, Feng et al., 2020], providing an accurate estimation of the true risk $R(\boldsymbol{g})$, i.e., $R(\boldsymbol{g}) = \bar{R}(\boldsymbol{g})$. Therefore, $\bar{R}(\boldsymbol{g})$ is referred as a URE of $R(\boldsymbol{g})$.

## 4 Complementary Label Distribution

Existing URE-based methods in MLCLL recover relevant labels and construct corresponding classifiers from complementary labels by making reasonable assumptions about the distribution of complementary labels. As a result, the complementary label distribution plays a crucial role in the modeling process. However, current assumption and theorem have certain limitations. Therefore, a systematic discussion and reasonable extension of them will be conducted in this section. We start from the uniform assumption and the existing URE derived from it.

**Assumption 4.1.** [Ishida et al., 2017, Gao et al., 2023] Uniform Distribution Assumption:

$$p(k = \bar{y} | \boldsymbol{x}_i) = \begin{cases} \frac{1}{K - |Y_i|}, & \text{if } k \notin Y_i, \\ 0, & \text{if } k \in Y_i. \end{cases} \tag{3}$$

Assumption 4.1 provides information that complementary labels are uniformly selected from the label space excluding relevant labels. This implies that each irrelevant label has an equal probability of being chosen as a complementary label by annotators. Based on this distribution, the contribution of $p(\boldsymbol{x}, Y)$ to $p(\boldsymbol{x}, \bar{y})$ is also uniform. As a result, a URE can be easily derived according to Assumption 4.1, which is shown as follows.

**Theorem 4.2.** *[Gao et al., 2023] URE under the uniform distribution: With $p(k = \bar{y}|\boldsymbol{x})$ defined in Assumption 4.1 and $R(\boldsymbol{g})$ defined in Eq. (1), the equality $\bar{R}(\boldsymbol{g}) = R_u(\boldsymbol{g})$ holds, where*

$$R_u(\boldsymbol{g}) = \mathbb{E}_{p(\boldsymbol{x}, \bar{y})}\left[\frac{1}{2^{K-1} - 1}\sum_{Y \subseteq \mathcal{Y}, \bar{y} \neq Y} \mathcal{L}(\boldsymbol{g}, Y)\right]. \tag{4}$$

However, in the real world, a uniform distribution may hardly cover actual scenarios. Some label combinations may be more likely to occur than others. Labels with a higher correlation to the relevant labels are closer to relevant labels and less likely to be chosen as complementary labels [Gao et al., 2024]. For example, when the relevant label is *water*, annotators are more likely to choose *desert* (a low co-occurrence label) rather than *lake* (a high co-occurrence label) as the complementary label. At the same time, *lake* is more likely to be closer to the relevant label compared to *desert*.

Therefore, we extend complementary labels from a uniform distribution to a biased distribution. Let matrix $\boldsymbol{L} = [l]_{|\mathcal{Y}| \times K}$ represents the correlation matrix, where $\boldsymbol{L}$ has $|\mathcal{Y}| = 2^K - 2$ rows, with each row corresponding to a label set $Y \in \mathcal{Y}$. The element $l_{Yk}$ represents the correlation between the label set $Y$ and the $k$-th label. The closer $Y$ is to the $k$-th label, the larger $l_{Yk}$ is. Note that $\boldsymbol{L}$ is used only for inference and does not appear in later computations.

**Assumption 4.3.** Biased Distribution Assumption:

$$p(k = \bar{y}|\boldsymbol{x}) = \begin{cases} \frac{z}{l_{Yk}}, & \text{if } k \notin Y, \\ 0, & \text{if } k \in Y, \end{cases} \tag{5}$$

where $z = \frac{1}{\sum_{k=1, k \notin Y}^{K} \frac{1}{l_{Yk}}}$, $l_{Yk} \propto p(k \in Y|\boldsymbol{x})$.

Assumption 4.3 summarizes the condition probabilities of a biased complementary label distribution. Here, $z$ in Eq. (5) is the normalization factor, ensuring that $p(k = \bar{y}|\boldsymbol{x})$ forms a valid probability distribution. The correlation $l_{Yk}$ in Eq. (5) is proportional to $p(k \in Y|\boldsymbol{x})$, meaning the correlation between label $k$ and set $Y$ is proportional to the conditional probability that label $k$ belongs to set $Y$, given $\boldsymbol{x}$.

Besides, the design of previous work is generally based on Label-Dependent Assumption, that is: The complementary label $\bar{y}$ is independent of the features $\boldsymbol{x}$ conditioned on the relevant label set $Y$, i.e., $p(\bar{y}|Y) = p(\bar{y}|\boldsymbol{x}, Y)$ [Ishida et al., 2017, 2019, Gao et al., 2023]. This assumption does not adequately encompass real-world scenarios, as annotators subconsciously select labels that are not too similar based on the instance's features, rather than the relevant labels, in the process of choosing complementary labels. Therefore, we adopt a more realistic Instance-dependent Assumption, which has been widely used in other weakly supervised scenarios [Xia et al., 2020, Chen et al., 2021, Kou et al., 2023].

**Assumption 4.4.** Instance-Dependent Assumption: Given an instance $\boldsymbol{x}$, the complementary label $\bar{y}$ is independent of $Y$, i.e. $p(\bar{y}|\boldsymbol{x}, Y) = p(\bar{y}|\boldsymbol{x})$.

Assumption 4.4 is a fundamental premise regarding the relationship between complementary labels and instances, positing that the selection of a complementary label depends on the instance. In some scenarios—such as an image containing multiple animals—annotators may struggle to exclude all relevant categories based solely on features. However, our assumption is motivated by the observation that annotators often eliminate obviously irrelevant labels by inspecting the input instance (e.g., excluding "building" from an image showing animals), without needing to infer the full set of relevant labels. This behavior supports modeling $\bar{y}$ as conditionally independent of $Y$ given $\boldsymbol{x}$. We adopt this assumption as a tractable approximation that reflects limited annotator under biased complementary label distributions.

Subsequently, we investigate the URE under the Biased Distribution Assumption. To simplify the computation, we first express the probability distribution in Assumption 4.4 in matrix form as a bias

transition matrix, denoted by $\bar{\boldsymbol{L}} = [\bar{l}_{Yk}]_{|\mathcal{Y}| \times K} \in \mathbb{R}^{(2^K - 2) \times K}$. Here, $\bar{l}_{Yk} = p(k = \bar{y}|\boldsymbol{x})$ represents the probability of label $k$ being selected as the complementary label when the relevant label set for $\boldsymbol{x}$ is $Y$. Theorem 4.5 then provides the URE derived from the biased complementary labels.

**Theorem 4.5.** *URE under biased distribution: Given $p(k = \bar{y}|\boldsymbol{x})$ and $R(\boldsymbol{g})$, the equality $R(\boldsymbol{g}) = \bar{R}(\boldsymbol{g}) = R_b(\boldsymbol{g})$ holds when*

$$R_b(\boldsymbol{g}) = \mathbb{E}_{p(\boldsymbol{x}, \bar{y})} \left[ \sum_{Y \subseteq \mathcal{Y}, \bar{y} \notin Y} \bar{l}^+_{Y\bar{y}} \mathcal{L}(\boldsymbol{g}, Y) \right], \tag{6}$$

*where $\bar{l}^+_{Y\bar{y}}$ belongs to the matrix $\bar{\boldsymbol{L}}^+ = [\bar{l}^+]_{|\mathcal{Y}| \times K}$, and $\bar{\boldsymbol{L}}^+$ is the Moore-Penrose pseudoinverse of $\bar{\boldsymbol{L}}^T$.*

The proof is stated in Appendix B. Compared to the URE under uniform distribution, the construction of URE (Eq. (6)) under biased distribution is heavily dependent on the specific characteristics of the distribution. The uniform assumption ensures that each complementary label contributes equally to the distribution of the relevant labels. However, once there is a bias in the contributions, $p(\boldsymbol{x}, \bar{y})$ is no longer uniform, which inevitably causes the URE to change. Therefore, the URE derived under the uniform assumption can no longer accurately estimate the expected risk once the distribution of complementary labels changes. In other words, it becomes ineffective under the biased assumption.

# 5 The Proposed Framework

## 5.1 Complementary Ranking Loss

In addition to the lack of universality across different distributions, URE-based methods do not take label correlations into account, thus losing important information needed to solve the MLL problem. Since complementary labels are known to be irrelevant, while non-complementary labels may include relevant ones, enforcing lower scores for complementary labels may help distinguish likely-relevant labels. This intuitive motivation naturally aligns with the goal of ranking loss, making it a suitable choice for integration into MLCLL. By incorporating label correlations without significantly increasing computational complexity, ranking loss has proven to be an effective tool for capturing pairwise label correlations [Zhang and Fang, 2020, Zhang et al., 2018, Fürnkranz et al., 2008]. In MLL, the traditional ranking loss is defined as

$$\mathcal{L}(\boldsymbol{g}(\boldsymbol{x}), Y) = \sum_{k=1, k \in Y}^{K} \sum_{j=1, j \notin Y}^{K} \mathbb{I}[\boldsymbol{g}_k(\boldsymbol{x}) < \boldsymbol{g}_j(\boldsymbol{x})],$$

where $\mathbb{I}(\cdot)$ is the indicator function, which outputs 1 when the condition holds and otherwise 0. Inspired by the complementary 0-1 loss [Chou et al., 2020], we propose the **complementary ranking loss** for MLCLL:

$$\bar{\mathcal{L}}(\boldsymbol{g}(\boldsymbol{x}), \bar{y}) = \sum_{k=1, k \neq \bar{y}}^{K} \mathbb{I}[\boldsymbol{g}_k(\boldsymbol{x}) < \boldsymbol{g}_{\bar{y}}(\boldsymbol{x})].$$

Unlike URE-based methods, the complementary ranking loss does not rely on any assumption regarding complementary labels. Instead, complementary ranking loss directly compares the predicted probabilities between different labels. Similar to ranking loss in supervised MLL, it penalizes cases where the complementary label predicted probability is higher than that of a non-complementary label, because a complementary label is certainly not a relevant label, while a non-complementary label may be either relevant or irrelevant. Therefore, it is generally reasonable to assign lower scores to complementary labels, as they are less likely to be relevant compared to non-complementary labels. The rationale behind this design will be formally justified in the next subsection.

However, directly optimizing the complementary ranking loss is challenging, as it is typically NP-hard due to its non-convexity and discontinuity. Thus, a convex surrogate loss can be introduced to facilitate more efficient optimization, a common method in ranking loss methods:

$$\bar{\mathcal{L}}(\boldsymbol{g}(\boldsymbol{x}), \bar{y}) = \sum_{k=1, k \neq \bar{y}}^{K} \ell(g_{\bar{y}}(\boldsymbol{x}) - g_k(\boldsymbol{x})),$$

---

**Algorithm 1** ComRank Algorithm

---

**Input:**
$\bar{D}$: the complementary-label training set $\{(\boldsymbol{x}_i, \bar{\boldsymbol{y}_i})\}_{i=1}^n$
$\boldsymbol{\theta}$: the initial parameters of classifier $\boldsymbol{g}$
$T$: the number of epochs
$\mathcal{A}$: an external stochastic optimization algorithm
**Output:**
$\boldsymbol{g}$: learned multi-label classifier
**Training Routine**
1: **for** $t = 1$ to $T$ **do**
2:     Let $\mathcal{L}$ be the risk,
3:         $\mathcal{L} = \frac{1}{n} \sum_{i=1}^n \{\bar{\mathcal{L}}_{\mathrm{CR}}(\boldsymbol{g}(\boldsymbol{x}_i), \bar{y}_i)\}$
4:     Set gradients $-\nabla_{\boldsymbol{\theta}} \mathcal{L}$
5:     Update $\boldsymbol{\theta}$ by $\mathcal{A}$ with $-\nabla_{\boldsymbol{\theta}} \mathcal{L}$
6: **end for**

---

where $\ell$ represents the surrogate loss. In this paper, we propose a complementary ranking loss framework named **ComR**ank, using an exponential function as the surrogate loss, shown as follows. The algorithm of applying ComRank can be referred from Algorithm 1.

$$\bar{\mathcal{L}}_{\mathrm{CR}}(\boldsymbol{g}(\boldsymbol{x}), \bar{y}) = \sum_{k=1, k \neq \bar{y}}^{K} \exp(g_{\bar{y}}(\boldsymbol{x}) - g_k(\boldsymbol{x})).$$

## 5.2 Bayes Consistency for ComRank

Bayes consistency is a desirable property of a loss function, ensuring that minimizing the expected loss leads to the Bayes prediction rule [Li et al., 2017, Cheng et al., 2010]. We say that a loss has Bayes consistency if it leads $\boldsymbol{g}$ to follow the Bayes prediction rule:

$$\boldsymbol{g}_k^*(\boldsymbol{x}) = p(k \in Y | \boldsymbol{x}).$$

In contrast, URE only guarantees that the expected risk of MLCLL aligns with that of fully supervised MLL, without directly evaluating the classification results. Therefore, Bayes consistency is a more rigorous criterion than URE.

To verify the rationality of ComRank, it's necessary to demonstrate ComRank's theoretical soundness by establishing Bayes consistency. The analysis begins with the following lemma.

**Lemma 5.1.** *Under $\bar{\mathcal{L}}_{\mathrm{CR}}(\boldsymbol{g}(\boldsymbol{x}), \bar{y})$, $\bar{\boldsymbol{g}}^k(\boldsymbol{x}) \geq \bar{\boldsymbol{g}}^j(\boldsymbol{x})$ if and only if $p(k = \bar{y} | \boldsymbol{x}) \leq p(j = \bar{y} | \boldsymbol{x})$.*

The proof can be found in Appendix C. The conclusion of Lemma 5.1 is achieved through risk minimization, which is a fundamental result that applies to all distributional scenarios of MLCLL. Moreover, this reflects the relationship between the predictive probabilities given by $\bar{\mathcal{L}}_{\mathrm{CR}}(\boldsymbol{g}(\boldsymbol{x}), \bar{y})$ and the probability of becoming a complementary label. With Lemma 5.1, we can demonstrate that $\bar{\mathcal{L}}_{\mathrm{CR}}(\boldsymbol{g}(\boldsymbol{x}), \bar{y})$ exhibits Bayes consistency under both the uniform distribution (Assumption 4.1) and the biased distribution (Assumption 4.3).

**Theorem 5.2.** *Bayes Consistency for ComRank: For both uniform and biased complementary label distributions, $\bar{\boldsymbol{g}}^k(\boldsymbol{x}) \geq \bar{\boldsymbol{g}}^j(\boldsymbol{x})$ holds under $\bar{\mathcal{L}}_{\mathrm{CR}}(\boldsymbol{g}(\boldsymbol{x}), \bar{y})$ if and only if $p(k \in Y | \boldsymbol{x}) \geq p(j \in Y | \boldsymbol{x})$.*

The proof is stated in Appendix D. Theorem 5.2 shows that ComRank establishes a theoretical connection whereby the ranking between complementary and non-complementary labels can be transferred to the ranking between irrelevant and relevant labels. Next, we will provide experimental results to support its performance.

Table 1: *Average Precision* (mean±std) on the training data with uniform complementary labels. The best performance of each dataset is shown in **boldface**, where ↓ / ↑ indicates that smaller/larger values of metrics are better performance.

| Methods | L-UW | CCMN | PMLMD | PARD | MAE | GDF | $R_u(\boldsymbol{g})$ | ComRank |
|---|---|---|---|---|---|---|---|---|
| scene | .395±.016 | .458±.019 | .441±.043 | .740±.020 | .432±.019 | .759±.012 | .734±.014 | **.780±.010** |
| yeast | .685±.018 | .646±.017 | .695±.023 | .608±.118 | .698±.018 | .712±.019 | .679±.016 | **.715±.019** |
| enron | .375±.037 | .337±.042 | .537±.092 | .444±.035 | .427±.047 | .620±.067 | .411±.038 | **.634±.068** |
| rcv1-s1 | .445±.020 | .409±.028 | .348±.030 | .468±.089 | .427±.029 | .471±.057 | .363±.019 | **.491±.118** |
| bibtex | .237±.009 | .259±.025 | .280±.044 | .447±.020 | .287±.009 | .614±.017 | .413±.016 | **.658±.014** |
| bookmark | .506±.009 | .397±.028 | .181±.005 | .549±.010 | .506±.007 | .619±.006 | .512±.007 | **.628±.005** |
| nuswideBoW | .451±.010 | .431±.014 | .457±.025 | .457±.057 | .466±.011 | .553±.008 | **.595±.008** | .585±.010 |

Table 2: Summary of pairwise t-test for ComRank against other comparing approaches at 0.05 significance level on uniform datasets, showing in form of Win/Tie/Loss.

| ComRank against | L-UW | CCMN | PMLMD | PARD | MAE | GDF | $R_u(\boldsymbol{g})$ | in total |
|---|---|---|---|---|---|---|---|---|
| *One Error* | 5/2/0 | 5/2/0 | 6/1/0 | 5/2/0 | 5/2/0 | 4/3/0 | 6/1/0 | **36/13/0** |
| *Coverage* | 6/1/0 | 7/0/0 | 5/2/0 | 6/1/0 | 6/1/0 | 4/3/0 | 6/1/0 | **40/9/0** |
| *Ranking Loss* | 6/1/0 | 7/0/0 | 6/1/0 | 6/1/0 | 6/1/0 | 4/3/0 | 6/1/0 | **41/8/0** |
| *Average Precision* | 6/1/0 | 6/1/0 | 6/1/0 | 6/1/0 | 5/2/0 | 4/3/0 | 6/0/1 | **39/9/1** |
| in total | **23/5/0** | **25/3/0** | **23/5/0** | **23/5/0** | **22/6/0** | **16/12/0** | **24/0/1** | **156/40/0** |

# 6 Experiments

## 6.1 Experimental Setup

**Datasets.** To fully verify the effectiveness of ComRank, we select seven multi-label datasets for experiments[2]. The range of their data sizes is from 1702 to 269648, and data domains include text, biology, and images. Their details can be referred from Appendix A. We unify data preprocessing on these datasets. To comprehensively illustrate the experimental impact of the single complementary label, in line with previous studies [Gao et al., 2023, 2024, Hang and Zhang, 2024], for datasets with label space larger than 50, we extract the 15 most frequently occurring labels and delete instances that did not appear with these labels.

**Data Processing.** We use uniform and biased complementary labels to conduct experiments. Specifically, 1) *Uniform complementary labels*: Each instance $\boldsymbol{x}_i$ is equipped with a complementary label randomly selected from $\{\mathcal{Y} - Y_i\}$, where irrelevant labels have an equal probability of being chosen; 2) *Biased complementary labels*: The selection of the complementary label for $\boldsymbol{x}_i$ follows a biased rule: based on co-occurrence rates computed from the original dataset, labels with lower co-occurrence rates are more likely to be selected. The model is trained on data annotated with complementary labels, while the test data is labeled with relevant labels.

**Baselines.** ComRank is compared with seven recent competitive methods, including one CLL method: L-UW; one MLL method: CCMN; two PML methods: PMLMD and PARD; and three MLCLL methods: MAE, GDF and $R_u(\boldsymbol{g})$, which are shown in the following details:

- L-UW [Gao and Zhang, 2021]: A CLL method that incorporates a weighted loss based on empirical risk to enhance the prediction gap between potential relevant labels and complementary labels.

- CCMN [Xie and Huang, 2023]: A MLL method that leverages class-conditional multi-label noise for learning, constructing two unbiased estimators.

- PMLMD [Xie et al., 2021]: A PML method in the form of ranking loss, specially equipped with weight and meta-disambiguation to figure out candidate labels in partial multi-label learning label sets.

---

[2]Publicly available at https://mulan.sourceforge.net/datasets-mlc.html.

Table 3: *Average Precision* (mean±std) on the training data with biased complementary labels. The best performance of each dataset is shown in **boldface**, where ↓ / ↑ indicates that smaller/larger values of metrics are better performance.

| Methods | L-UW | CCMN | PMLMD | PARD | MAE | GDF | $R_u(\boldsymbol{g})$ | ComRank |
|---|---|---|---|---|---|---|---|---|
| scene | .398±.021 | .461±.020 | .445±.032 | .691±.021 | .424±.018 | .729±.020 | .716±.026 | **.751±.015** |
| yeast | .691±.018 | .650±.032 | .702±.025 | .661±.039 | .703±.018 | .714±.019 | .691±.014 | **.717±.019** |
| enron | .373±.038 | .356±.052 | .571±.065 | .432±.055 | .439±.050 | .610±.078 | .415±.041 | **.626±.080** |
| rcv1-s1 | .450±.029 | .386±.028 | .341±.034 | .485±.098 | .422±.042 | .521±.021 | .413±.044 | **.542±.099** |
| bibtex | .234±.011 | .268±.031 | .329±.104 | .446±.017 | .285±.010 | .627±.019 | .414±.018 | **.665±.021** |
| bookmark | .525±.012 | .389±.024 | .180±.005 | .557±.009 | .518±.008 | .629±.006 | .517±.013 | **.647±.006** |
| nuswideBoW | .439±.011 | .402±.028 | .457±.023 | .496±.014 | .454±.012 | .556±.012 | **.594±.008** | .589±.010 |

Table 4: Summary of pairwise t-test for ComRank against other comparing approaches at 0.05 significance level on biased datasets, showing in form of Win/Tie/Loss.

| ComRank against | L-UW | CCMN | PMLMD | PARD | MAE | GDF | $R_u(\boldsymbol{g})$ | in total |
|---|---|---|---|---|---|---|---|---|
| *One Error* | 6/1/0 | 6/1/0 | 6/1/0 | 5/2/0 | 6/1/0 | 4/3/0 | 5/2/0 | **38/11/0** |
| *Coverage* | 6/1/0 | 7/0/0 | 5/2/0 | 7/0/0 | 7/0/0 | 4/3/0 | 7/0/0 | **43/6/0** |
| *Ranking Loss* | 6/1/0 | 7/0/0 | 5/2/0 | 7/0/0 | 7/0/0 | 1/6/0 | 7/0/0 | **40/9/0** |
| *Average Precision* | 7/0/0 | 7/0/0 | 5/2/0 | 6/1/0 | 6/1/0 | 4/3/0 | 6/1/0 | **41/8/0** |
| in total | **25/3/0** | **27/1/0** | **21/7/0** | **25/3/0** | **26/2/0** | **13/14/0** | **25/3/0** | **162/34/0** |

- PARD [Hang and Zhang, 2024]: A PML method based on a probabilistic graphical model, designed to infer potential ground-truth label information by modeling the generation process of partial multi-label data.

- MAE [Gao et al., 2023]: A MLCLL method that leverages MAE loss within the URE framework for learning.

- GDF [Gao et al., 2023]: A MLCLL method that utilizes a gradient descent-friendly loss based on URE.

- $R_u(\boldsymbol{g})$ [Gao et al., 2023]: The MLCLL loss function derived from $R_u(\boldsymbol{g})$ in Eq. (4), the URE based on uniform complementary labels.

**Evaluation Metrics.** We evaluate performance using four common MLL metrics: *Ranking Loss*, *Coverage*, *One Error* and *Average Precision*. Their details can be referred from Zhang and Zhou [2014]. Notably, the metric of *Ranking Loss* evaluates the fraction of reversely ordered label pairs, i.e., an irrelevant label is ranked higher than a relevant label, which is different from our method.

**Implementation Details.** Our experiments are conducted using PyTorch [Paszke et al., 2019] and implement on an NVIDIA TITAN RTX. To ensure fair comparisons, a linear model is applied to all datasets. For statistical analysis, we employ ten-fold cross-validation, where the dataset is randomly divided into ten subsets. The results are reported as the mean and *standard deviation* (std) of the metric. The weight decay was set to $1e-3$, and the learning rate was selected from $\{1e-3, 1e-2, 1e-1\}$. It is multiplied by 0.1 when the iteration count reaches 100 and 150 [Wang et al., 2021]. The training epochs for all datasets are 200. These settings were kept consistent across all methods.

## 6.2 Comparison on Uniform Complementary Labels

To evaluate the effectiveness of our method under uniform complementary label situation, we use uniform complementary-labeled data to train.

**Results.** Table 1 reports the results for *Average Precision*, while the results for *Coverage*, *One Error*, and *Ranking Loss* are provided in Table 8 of Appendix E due to space limitations. As shown, ComRank achieves significant improvements across all datasets. Among the 42 dataset-metric combinations, ComRank achieves the best performance in 39 cases. The most notable improvement occurs on the bibtex dataset, where *Average Precision* increases from 0.237 (under the L-UW method)

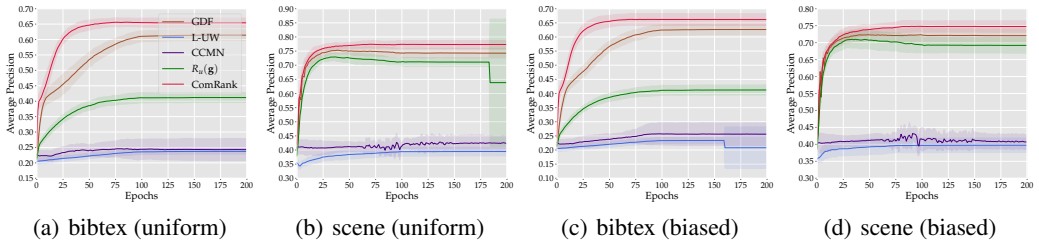

| (a) bibtex (uniform) | (b) scene (uniform) | (c) bibtex (biased) | (d) scene (biased) |

Figure 1: *Average Precision* on datasets with uniform or biased complementary labels. Dark lines show the mean of testing results, where light shadows correspond to the std.

Table 5: The running time (in $10^2$ seconds) of methods with multiple uniform complementary labels.

| Dataset | #Label Classes | CCMN | L-UW | GDF | MAE | $R_u(g)$ | ComRank |
|---------|---------------|------|------|-----|-----|----------|---------|
| enron | 53 | 2.94 | 3.22 | 3.79 | 3.72 | 3.14 | 3.15 |
| rcv1-s1 | 101 | 4.13 | 3.67 | 4.12 | 3.56 | 3.76 | 3.64 |
| bibtex | 159 | 5.11 | 3.76 | 4.07 | 3.69 | 3.89 | 3.91 |
| bookmark | 208 | 33.25 | 17.11 | 17.18 | 18.68 | 17.23 | 17.19 |
| avg | - | 11.36 | 6.94 | 7.29 | 7.41 | 7.01 | 6.97 |

to 0.658, clearly demonstrating ComRank's strong learning capability. Additionally, compared to CLL methods, ComRank reduces the *One Error* from 0.73 to 0.308 on the enron dataset, highlighting its adaptability to MLL in the complementary label setting.

**Statistical Tests.** Table 2 presents the pairwise $t$-test results of ComRank against seven methods at a 0.05 significance level across four metrics. For each pairwise comparison, we use the $t$-test to determine statistical significance. If ComRank significantly outperforms the baseline, we add 1 to the win count; if ComRank is significantly worse, we add 1 to the loss count; otherwise, we add 1 to the tie count. The final results are reported in the format of win/tie/loss. ComRank consistently outperforms the baselines statistically, achieving 23/5/0 against L-UW, 25/3/0 against CCMN, and showing similar strong performance against other methods. Particularly, ComRank demonstrates dominance in metrics *One Error*, *Coverage* and *Ranking Loss*, where it achieves no losses across almost all baselines. These results highlight the effectiveness of ComRank.

**Convergence Analysis on the Uniform Distribution.** Figure 1 shows the epoch situation of *Average Precision* for compared methods and ComRank in bibtex and scene datasets, and similar tendency also shows on other datasets. Remaining metrics are in Figure 2 of Appendix G. As shown in the figure, ComRank demonstrates the best performance among all methods, exhibiting the fastest and most substantial improvement in *Average Precision*. Notably, GDF, $R_u(\mathbf{g})$, and CCMN show slight instability, with oscillations emerging midway or toward the end of training. In contrast, ComRank maintains high stability throughout the optimization process, highlighting its effectiveness in gradient descent when complementary labels are selected uniformly.

### 6.3 Comparison on Biased Complementary Labels

To assess the effectiveness of our method under biased distribution, we train from biased complementary-labeled data generated by the co-occurrence rates of relevant labels.

**Results.** Table 3 shows the results of *Average Precision*, while *Coverage*, *One Error* and *Ranking Loss* results are in Table 9 of Appendix F due to page limitation. Similar to the uniform setting, ComRank demonstrates significant improvements across all datasets. Out of the seven datasets, it achieves the best performance across all metrics on five. Among them, on the scene dataset, *One Error* is reduced dramatically from 0.852 (under L-UW) to 0.402. Compared to these methods, especially URE-based methods such as $R_u(\boldsymbol{g})$ and GDF, ComRank continues to show superior performance by reducing *Coverage* on scene from 0.422 (under $R_u(\boldsymbol{g})$) to 0.130, and improving *Average Precision* on bibtex from 0.614 (under GDF) to 0.658. These results clearly demonstrate ComRank's advantage in handling biased complementary label scenarios.

Table 6: Comparison of Frobenius norm distances for label correlation preservation.

| Datasets | CCMN | L-UW | GDF | MAE | $R_u(g)$ | ComRank |
|---|---|---|---|---|---|---|
| scene | 6.44 | 6.38 | 1.83 | 6.42 | 1.36 | 1.82 |
| yeast | 4.45 | 4.68 | 5.79 | 5.92 | 3.70 | 6.34 |
| enron | 14.92 | 12.88 | 14.33 | 14.84 | 3.33 | 3.65 |
| rcv1-s1 | 12.42 | 15.06 | 11.32 | 15.04 | 3.20 | 4.95 |
| bibtex | 15.18 | 14.98 | 13.69 | 15.22 | 1.80 | 2.71 |
| bookmark | 13.38 | 15.09 | 11.38 | 15.09 | 1.27 | 11.26 |
| nuswideBoW | 14.52 | 14.45 | 9.22 | 14.58 | 4.93 | 8.96 |

**Statistical Tests.** Table 4 displays the pairwise t-test results of ComRank compared to various baselines on biased datasets. In particular, ComRank demonstrates superiority for all metrics with no losses in all cases. It is worth noting that, ComRank achieves near-perfect performance compared with $R_u(\boldsymbol{g})$ (25/3/0), underscoring its adaptability across datasets and biased complementary label scenarios.

**Convergence Analysis on the Biased Distribution.** Figure 1 shows the epoch situation of *Average Precision* for compared methods and ComRank on various datasets. Figures of *Coverage*, *Ranking Loss* and *One Error* are in Figure 3 of Appendix G. Also, the curve of ComRank shows the best performance among all methods, and remains highly stable, demonstrating its effectiveness in the gradient descent process, especially for biased complementary labels.

## 6.4 Further Analysis.

**Complexity Analysis.** With a single complementary label per instance, the proposed ComRank method has a computational complexity of $O(n(K-1))$. Although the complexity grows with more complementary labels, our implementation avoids the high cost of pairwise comparisons by leveraging matrix operations with masking, ensuring efficiency even for large label spaces. To empirically assess scalability, we report the running time on datasets where each instance has $K/2$ uniform complementary labels (Table 5). ComRank achieves comparable or faster speeds than most baselines, demonstrating its scalability.

**Label Correlation Preservation.** We evaluate each method's ability to preserve label correlations by comparing the Pearson correlation matrices of the test labels and the predicted scores, under uniform complementary labels. Their difference, measured by the Frobenius norm distance (lower is better), is reported in Table 6. ComRank achieves superior correlation preservation over most baselines, demonstrating that its ranking-based design effectively retains meaningful label dependencies.

**Surrogate Loss Ablation.** The table 10 in Appendix H reports the *Average Precision* from an ablation study on different surrogate losses under uniform complementary labels. We compared for log loss, sigmoid loss, softmax loss and ComRank (with exponential loss). Their details can be referred to from Appendix H. Comrank achieves competitive performance in most cases, validating its effectiveness and Bayes consistency.

# 7 Conclusion

In this paper, we theoretically analyze the limitations of URE, revealing that its reliance on distributional assumptions restricts its effectiveness to scenarios with uniformly selected complementary labels. Under biased complementary labels, URE struggles to provide unbiased risk estimation and fails to capture inter-label relationships. To address these issues, we propose ComRank, a complementary ranking loss framework that is theoretically justified under both uniform and biased complementary label settings. Our risk minimization analysis demonstrates that ComRank has Bayes consistency in both cases. Experimental results further validate its remarkable stability and effectiveness in learning. A current limitation of this work is that ComRank is based on multiple assumptions, including distributional assumptions and independence assumptions. In the future, we hope to extend it to more general scenarios.

## Acknowledgements

The authors wish to thank the anonymous reviewers for their helpful comments and suggestions. This work was supported by the National Science Foundation of China (62225602, 624B2042). MX is supported by the Australian Research Council (DE230101116). We thank the Big Data Center of Southeast University for providing the facility support on the numerical calculations in this paper.

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

# Appendix

## A Details of Datasets

Table 7 describes the datasets used in the experiments of this paper, where #Instance represents the number of samples, #Features represents the sample feature dimension, #Label Classes represents the label space, #Cardinality represents the average number of relevant labels of the sample, and #Domain represents the data type.

Table 7: Characteristics of datasets.

| Datasets | #Instances | #Features | #Label Classes | #Cardinality | #Domain |
|---|---|---|---|---|---|
| scene | 2407 | 294 | 6 | 1.07 | images |
| yeast | 2417 | 103 | 14 | 4.23 | biology |
| enron | 1702 | 1001 | 53 | 3.38 | Text |
| rcv1-s1 | 5815 | 944 | 101 | 2.88 | Text |
| bibtex | 7365 | 1836 | 159 | 2.40 | images |
| bookmark | 87856 | 2150 | 208 | 1.25 | Text |
| nuswideBoW | 269648 | 500 | 81 | 1.87 | images |

## B The Proof of Theorem 4.5

**Theorem 4.5.** *URE under biased distribution: Given $p(k = \bar{y}|\boldsymbol{x})$ and $R(\boldsymbol{g})$, the equality $R(\boldsymbol{g}) = \bar{R}(\boldsymbol{g}) = R_b(\boldsymbol{g})$ holds when*

$$R_b(\boldsymbol{g}) = \mathbb{E}_{p(\boldsymbol{x},\bar{y})} \left[ \sum_{Y \subseteq \mathcal{Y}, \bar{y} \neq Y} \bar{l}^+_{Y\bar{y}} \mathcal{L}(\boldsymbol{g}, Y) \right], \tag{7}$$

*where $\bar{l}^+_{Y\bar{y}}$ belongs to the matrix $\bar{\boldsymbol{L}}^+ = [\bar{l}^+]_{|\mathcal{Y}| \times K}$, and $\bar{\boldsymbol{L}}^+$ is the Moore-Penrose pseudoinverse of $\bar{\boldsymbol{L}}^T$.*

*Proof.* According to Assumption 4.3 and Assumption 4.4, for $\bar{y} \notin Y$, $p(\bar{y}|\boldsymbol{x}, Y) = \frac{z}{l_{Y\bar{y}}}$. Therefore,

$$p(\boldsymbol{x}, \bar{y}) = \sum_{Y \subseteq \mathcal{Y}, \bar{y} \notin Y} p(\boldsymbol{x}, Y, \bar{y}) = \sum_{Y \subseteq \mathcal{Y}, \bar{y} \notin Y} p(\boldsymbol{x}, Y) p(\bar{y}|\boldsymbol{x}, Y)$$

$$= \sum_{Y \subseteq \mathcal{Y}, \bar{y} \notin Y} \frac{z}{l_{Y\bar{y}}} p(\boldsymbol{x}, Y).$$

Set $\bar{\boldsymbol{L}} = [\bar{l}]_{|\mathcal{Y}| \times K}$, where $\bar{l}_{Yk} = \begin{cases} \frac{z}{l_{Yk}}, & k \in Y \\ 0, & k \notin Y \end{cases}$. By expanding $p(\boldsymbol{x}, \bar{y})$ and $p(\boldsymbol{x}, Y)$ with respect to $\bar{y}$ and $Y$ as marginal probabilities, we can obtain:

$$\begin{bmatrix} p(\boldsymbol{x}, \bar{y} = 1) \\ \vdots \\ p(\boldsymbol{x}, \bar{y} = k) \\ \vdots \\ p(\boldsymbol{x}, \bar{y} = K) \end{bmatrix}_{K \times 1} = \bar{\boldsymbol{L}}^T \begin{bmatrix} p(\boldsymbol{x}, Y = Y_1) \\ p(\boldsymbol{x}, Y = Y_2) \\ \vdots \\ p(\boldsymbol{x}, Y = Y_{|\mathcal{Y}|}) \end{bmatrix}_{|\mathcal{Y}| \times 1}.$$

When $\bar{L}^T$ is full rank, there exists a matrix $\bar{L}^+ = [\bar{l}^+]_{|\mathcal{Y}|\times K}$, which is the Moore-Penrose pseudoinverse of $\bar{L}^T$, satisfies:

$$\bar{L}^+ \cdot \begin{bmatrix} p(\boldsymbol{x},\bar{y}=1) \\ \vdots \\ p(\boldsymbol{x},\bar{y}=k) \\ \vdots \\ p(\boldsymbol{x},\bar{y}=K) \end{bmatrix}_{K\times 1} = \begin{bmatrix} p(\boldsymbol{x},Y=Y_1) \\ p(\boldsymbol{x},Y=Y_2) \\ \vdots \\ p(\boldsymbol{x},Y=Y_{|\mathcal{Y}|}) \end{bmatrix}_{|\mathcal{Y}|\times 1} .$$

Therefore, we can have the relationship

$$p(\boldsymbol{x},Y) = \sum_{\bar{y}=1,\bar{y}\neq Y}^{K} \bar{l}^+_{Y\bar{y}} p(\boldsymbol{x},\bar{y}),$$

where $\bar{l}^+_{Y\bar{y}}$ is the $\bar{y}$-th column of the row corresponding to $Y$ in $\bar{L}^+$. Accordingly, the URE under Assumption 4.3 is:

$$\begin{aligned}
R_b(\boldsymbol{g}) &= \mathbb{E}_{p(\boldsymbol{x},Y)}\left[\mathcal{L}(\boldsymbol{g},Y)\right] \\
&= \int_{\mathcal{X}} \sum_{Y\subseteq\mathcal{Y}} \mathcal{L}(\boldsymbol{g},Y) p(\boldsymbol{x},Y) d\boldsymbol{x} \\
&= \int_{\mathcal{X}} \sum_{Y\subseteq\mathcal{Y}} \sum_{\bar{y}=1}^{K} \mathcal{L}(\boldsymbol{g},Y) \bar{l}^+_{Y\bar{y}} p(\boldsymbol{x},\bar{y}) d\boldsymbol{x} \\
&= \int_{\mathcal{X}} \sum_{\bar{y}=1}^{K} \sum_{Y\subseteq\mathcal{Y},\bar{y}\neq Y} \mathcal{L}(g,Y) \bar{l}^+_{Y\bar{y}} \bar{p}(\boldsymbol{x},\bar{y}) d\boldsymbol{x} \\
&= \mathbb{E}_{p(\boldsymbol{x},\bar{y})}\left[ \sum_{Y\subseteq\mathcal{Y},\bar{y}\neq Y} \bar{l}^+_{Y\bar{y}} \mathcal{L}(\boldsymbol{g},Y) \right].
\end{aligned}$$

$\square$

## C   The Proof of Lemma 5.1

**Lemma 5.1.** *Under* $\bar{\mathcal{L}}_{\mathrm{CR}}(\boldsymbol{g}(\boldsymbol{x}),\bar{y})$, $\bar{g}^k(\boldsymbol{x}) \geq \bar{g}^j(\boldsymbol{x})$ *if and only if* $p(k=\bar{y}|\boldsymbol{x}) \leq p(j=\bar{y}|\boldsymbol{x})$.

*Proof.* Let $\Delta Y_k = \{0,1,-1\} \in \mathbb{R}^{1\times K}$, where the $\bar{y}$-th position is 1, the $k$-th position is -1, and other positions are 0, then:

$$\bar{\mathcal{L}}_{\mathrm{CR}}(\boldsymbol{g}(\boldsymbol{x}),\bar{y}) = \sum_{k=1,k\neq\bar{y}}^{K} \exp(\boldsymbol{g}(\boldsymbol{x})\Delta Y_k^T).$$

Assume $\alpha_k = \exp(\boldsymbol{g}(\boldsymbol{x})\Delta Y_k^T)$, we can obtain

$$\begin{aligned}
\bar{R}(\boldsymbol{g}) &= \mathbb{E}_p(\boldsymbol{x},\bar{y})\left[\bar{\mathcal{L}}_{\mathrm{CR}}(\boldsymbol{g}(\boldsymbol{x}),\bar{y})\right] \\
&= \int_{\mathcal{X}} \int_{\mathcal{Y}} \bar{\mathcal{L}}_{\mathrm{CR}}(\boldsymbol{g},\bar{y}) p(\boldsymbol{x},\bar{y}) d\boldsymbol{x}\, d\bar{y} \\
&= \int_{\mathcal{X}} \sum_{\bar{y}\in\mathcal{Y}} \bar{\mathcal{L}}_{\mathrm{CR}}(\boldsymbol{g},\bar{y}) p(\boldsymbol{x},\bar{y}) d\boldsymbol{x} \\
&= \int_{\mathcal{X}} \sum_{\bar{y}\in\mathcal{Y}} \sum_{k=1,k\neq\bar{y}}^{K} \sum_{j=1,j=\bar{y}}^{K} \alpha_k\, p(\boldsymbol{x},\bar{y}) d\boldsymbol{x} \\
&= \int_{\mathcal{X}} \sum_{k=1}^{K} \sum_{j=1}^{K} \sum_{\bar{y}\in\mathcal{Y},k\neq\bar{y},j=\bar{y}} \alpha_k\, p(\boldsymbol{x},\bar{y}) d\boldsymbol{x}
\end{aligned}$$

$$= \int_{\mathcal{X}} \sum_{k=1}^{K} \sum_{j=1}^{K} \sum_{\bar{y} \in \mathcal{Y}, k \neq \bar{y}, j = \bar{y}} \alpha_k \, p(\bar{y}|\boldsymbol{x}) \, p(\boldsymbol{x}) \, d\boldsymbol{x}$$

$$= \int_{\mathcal{X}} \sum_{k=1}^{K} \sum_{j=1}^{K} p(k \neq \bar{y}, j = \bar{y}|\boldsymbol{x}) \, \alpha_k \, p(\boldsymbol{x}) \, d\boldsymbol{x}.$$

Since the minimization is performed with respect to $\boldsymbol{g}$, and $\boldsymbol{x}$ does not affect the minimization, $\bar{\boldsymbol{g}}^* = \arg\min_{\boldsymbol{g} \in G} \bar{R}(\boldsymbol{g})$ can be converted to

$$\bar{\boldsymbol{g}}'^* = \arg\min_{\boldsymbol{g} \in G} \bar{R}'(\boldsymbol{g}) = \sum_{k=1}^{K} \sum_{j=1}^{K} p(k \neq \bar{y}, j = \bar{y}|\boldsymbol{x})\alpha_k$$

When $\bar{R}'(\boldsymbol{g})$ reaches $\boldsymbol{g}'^*(\boldsymbol{x})$, $\bar{R}(\boldsymbol{g})$ can reach $\boldsymbol{g}^*(\boldsymbol{x})$. Let $\beta_k = p(k \neq \bar{y}, j = \bar{y}|\boldsymbol{x})$, thus the first-order derivative is

$$\frac{\partial \bar{R}'}{\partial \boldsymbol{g}} = \sum_{k=1}^{K} \sum_{j=1}^{K} \beta_k \Delta Y_k \alpha_k.$$

And the second derivative is

$$\frac{\partial^2 \bar{R}'}{\partial \boldsymbol{g}^2} = \sum_{k=1}^{K} \sum_{j=1}^{K} \beta_k \Delta Y_k^T \Delta Y_k \alpha_k \geq \boldsymbol{0}_{K \times K}.$$

Therefore when the first derivative is equal to 0, $\bar{R}'(\boldsymbol{g})$ has the minimum. Let $\frac{\partial \bar{R}'}{\partial \boldsymbol{g}} = \boldsymbol{0}$, we have

$$\sum_{k=1}^{K} \sum_{j=1}^{K} \beta_k \Delta Y_k \alpha_k = \boldsymbol{0}$$

$$\Rightarrow \sum_{k=1}^{K} \sum_{j=1}^{K} \beta_k \Delta Y_k \exp\left(\bar{\boldsymbol{g}}'^*(\boldsymbol{x}) \Delta Y_k^T\right) = \boldsymbol{0}$$

$$\Rightarrow \sum_{k=1}^{K} \sum_{j=1}^{K} \beta_k \Delta Y_k \exp\left(\bar{\boldsymbol{g}}_j'^*(\boldsymbol{x}) - \bar{\boldsymbol{g}}_k'^*(\boldsymbol{x})\right) = \boldsymbol{0}. \quad (8)$$

Therefore, the first-order derivative is $\boldsymbol{0}$ when

$$\exp\left(\bar{\boldsymbol{g}}_j'^*(x) - \bar{\boldsymbol{g}}_k'^*(\boldsymbol{x})\right) = \frac{p(k = \bar{y}, j \neq \bar{y}|\boldsymbol{x})}{p(k \neq \bar{y}, j = \bar{y}|\boldsymbol{x})},$$

because at this moment for each dimension in Eq. (8),

$$\sum_{j=1}^{K} p(k = \bar{y}, j \neq \bar{y}|\boldsymbol{x}) - \sum_{k=1}^{K} p(k = \bar{y}, j \neq \bar{y}|\boldsymbol{x}) = 0.$$

Then, we have

$$\bar{\boldsymbol{g}}_j'^*(\boldsymbol{x}) - \bar{\boldsymbol{g}}_k'^*(\boldsymbol{x}) = \log \frac{p(k = \bar{y}, j \neq \bar{y}|\boldsymbol{x})}{p(k \neq \bar{y}, j = \bar{y}|\boldsymbol{x})}, \quad \forall k, j \in \mathcal{Y}$$

$$\Rightarrow \bar{\boldsymbol{g}}_k'^*(\boldsymbol{x}) - \bar{\boldsymbol{g}}_j'^*(\boldsymbol{x}) = \log \frac{p(k \neq \bar{y}, j = \bar{y}|\boldsymbol{x})}{p(k = \bar{y}, j \neq \bar{y}|\boldsymbol{x})}, \quad \forall k, j \in \mathcal{Y}.$$

Therefore, $\bar{\boldsymbol{g}}_k'^*(\boldsymbol{x}) > \bar{\boldsymbol{g}}_j'^*(\boldsymbol{x})$ if and only if $p(k \neq \bar{y}, j = \bar{y}|\boldsymbol{x}) \geq p(k = \bar{y}, j \neq \bar{y}|\boldsymbol{x})$ holds. That is, $\bar{\boldsymbol{g}}_k^*(\boldsymbol{x}) > \bar{\boldsymbol{g}}_j^*(\boldsymbol{x})$ if and only if $p(k \neq \bar{y}, j = \bar{y}|\boldsymbol{x}) \geq p(k = \bar{y}, j \neq \bar{y}|\boldsymbol{x})$ holds. Then, we have

$$p(k \neq \bar{y}|j = \bar{y}, \boldsymbol{x})p(j = \bar{y}|\boldsymbol{x}) \geq p(j \neq \bar{y}|k = \bar{y}, \boldsymbol{x})p(k = \bar{y}|\boldsymbol{x})$$

$$\because p(k \neq \bar{y}|j = \bar{y}, \boldsymbol{x}) = p(k \neq \bar{y}|\boldsymbol{x}) - p(k \neq \bar{y}, j \neq \bar{y}|\boldsymbol{x})$$

$$\Rightarrow [p(k \neq \bar{y}|\boldsymbol{x}) - p(k \neq \bar{y}, j \neq \bar{y}|\boldsymbol{x})]p(j = \bar{y}|\boldsymbol{x}) \geq [p(j \neq \bar{y}|\boldsymbol{x}) - p(j \neq \bar{y}, k \neq \bar{y}|\boldsymbol{x})]p(k = \bar{y}|\boldsymbol{x})$$

$$\Rightarrow p(k \neq \bar{y}|\boldsymbol{x})p(j = \bar{y}|\boldsymbol{x}) - p(k \neq \bar{y}, j \neq \bar{y}|\boldsymbol{x})p(j = \bar{y}|\boldsymbol{x})$$
$$\geq p(j \neq \bar{y}|\boldsymbol{x})p(k = \bar{y}|\boldsymbol{x}) - p(j \neq \bar{y}, k \neq \bar{y}|\boldsymbol{x})p(k = \bar{y}|\boldsymbol{x})$$
$$\because p(j = \bar{y}|\boldsymbol{x}) = 1 - p(j \neq \bar{y}|\boldsymbol{x}), p(k = \bar{y}|\boldsymbol{x}) = 1 - p(k \neq \bar{y}|\boldsymbol{x})$$
$$\Rightarrow p(k \neq \bar{y}|\boldsymbol{x}) - p(k \neq \bar{y}|\boldsymbol{x})p(j \neq \bar{y}|\boldsymbol{x}) - p(k \neq \bar{y}, j \neq \bar{y}|\boldsymbol{x}) + p(k \neq \bar{y}, j \neq \bar{y}|\boldsymbol{x})p(j \neq \bar{y}|\boldsymbol{x})$$
$$\geq p(j \neq \bar{y}|\boldsymbol{x}) - p(j \neq \bar{y}|\boldsymbol{x})p(k \neq \bar{y}|\boldsymbol{x}) - p(j \neq \bar{y}, k \neq \bar{y}|\boldsymbol{x}) + p(j \neq \bar{y}, k \neq \bar{y}|\boldsymbol{x})p(k \neq \bar{y}|\boldsymbol{x})$$
$$\Rightarrow p(k \neq \bar{y}|\boldsymbol{x}) - p(j \neq \bar{y}|\boldsymbol{x}) \geq p(k \neq \bar{y}, j \neq \bar{y}|\boldsymbol{x})(p(k \neq \bar{y}|\boldsymbol{x}) - p(j \neq \bar{y}|\boldsymbol{x})).$$

This means if and only if $p(k \neq \bar{y}) \geq p(j \neq \bar{y}|\boldsymbol{x})$ does the inequality hold. Therefore, $\bar{g}_k^*(\boldsymbol{x}) \geq \bar{g}_j^*(\boldsymbol{x})$ if and only if $p(k \neq \bar{y}|\boldsymbol{x}) \geq p(j \neq \bar{y}|\boldsymbol{x})$ holds. $\qquad\square$

## D  The Proof of Theorem 5.2

**Theorem 5.2.** *Bayes Consistency for ComRank: For both uniform and biased complementary label distributions, $\bar{g}^k(\boldsymbol{x}) \geq \bar{g}^j(\boldsymbol{x})$ holds under $\bar{\mathcal{L}}_{\mathrm{CR}}(\boldsymbol{g}(\boldsymbol{x}), \bar{y})$ if and only if $p(k \in Y|\boldsymbol{x}) \geq p(j \in Y|\boldsymbol{x})$.*

**Theorem 5.2.1.** *Under $\bar{\mathcal{L}}_{\mathrm{CR}}(\boldsymbol{g}(\boldsymbol{x}), \bar{y})$, $\bar{g}^k(\boldsymbol{x}) \geq \bar{g}^j(\boldsymbol{x})$ if and only if $p(k \in Y|\boldsymbol{x}) \geq p(j \in Y|\boldsymbol{x})$, for Assumption 4.1.*

*Proof.* According to Lemma 5.1, $\bar{g}_k^*(\boldsymbol{x}) > \bar{g}_j^*(\boldsymbol{x})$ if and only if $p(k = \bar{y}|\boldsymbol{x}) \leq p(j = \bar{y}|\boldsymbol{x})$.

Based on Assumption 4.1, whenever $k \in \bar{y}$ and $j \notin \bar{y}$, it holds that $p(k = \bar{y}|\boldsymbol{x}) \leq p(j = \bar{y}|\boldsymbol{x})$.

Additionally, when $k \in \bar{y}$ and $j \in \bar{y}$, $p(k \in \bar{y}|\boldsymbol{x}) \geq p(j \in \bar{y}|\boldsymbol{x})$.

That is, $\bar{g}_k^*(\boldsymbol{x}) > \bar{g}_j^*(\boldsymbol{x})$ if and only if $p(k \in \bar{y}|\boldsymbol{x}) \geq p(j \in \bar{y}|\boldsymbol{x})$, which satisfies Bayes consistency. $\qquad\square$

**Theorem 5.2.2.** *Under $\bar{\mathcal{L}}_{\mathrm{CR}}(\boldsymbol{g}(\boldsymbol{x}), \bar{y})$, $\bar{g}^k(\boldsymbol{x}) \geq \bar{g}^j(\boldsymbol{x})$ if and only if $p(k \in Y|\boldsymbol{x}) \geq p(j \in Y|\boldsymbol{x})$, for Assumption 4.3.*

*Proof.* According to Lemma 5.1, $\bar{g}_k^*(\boldsymbol{x}) > \bar{g}_j^*(\boldsymbol{x})$ if and only if $p(k = \bar{y}|\boldsymbol{x}) \leq p(j = \bar{y}|\boldsymbol{x})$. According to Assumption 4.3, when $p(k = \bar{y}|\boldsymbol{x}) \leq p(j = \bar{y}|\boldsymbol{x})$, there are two possible cases:

1. $k \in Y, j \notin Y$: In this case, we must have $p(k \in Y|\boldsymbol{x}) \geq p(j \in Y|\boldsymbol{x})$.

That is, $\bar{g}_k^*(\boldsymbol{x}) \geq \bar{g}_j^*(\boldsymbol{x})$ if and only if $p(k \in Y|\boldsymbol{x}) \geq p(j \in Y|\boldsymbol{x})$, which satisfies Bayes consistency.

2. $k, j \notin Y$: Since $p(k = \bar{y}|\boldsymbol{x}) = \frac{z}{l_{Yk}}$, we obtain $\Rightarrow \frac{z}{l_{Yk}} \leq \frac{z}{l_{Yj}}$.

Since $z$ is same, we have $l_{Yk} \geq l_{Yj}$.

$\because l_{Yk} \propto p(k \in Y|\boldsymbol{x})$, we have $p(k \in Y|\boldsymbol{x}) \geq p(j \in Y|\boldsymbol{x})$.

That is, $\bar{g}_k^*(\boldsymbol{x}) > \bar{g}_j^*(\boldsymbol{x})$ if and only if $p(k \in Y|\boldsymbol{x}) \geq p(j \in Y|\boldsymbol{x})$, which satisfies Bayes consistency. $\qquad\square$

## E  Comparison on Uniform Complementary Labels

Table 8 shows the comparison of ComRank against multiple methods in *One Error*, *Ranking Loss* and *Coverage* under uniform complementary labels. As we can see, ComRank demonstrates strong performance across all methods and achieves superior results on most datasets.

## F  Comparison on Biased Complementary Labels

Table 9 shows the comparison of ComRank against multiple methods in *One Error*, *Ranking Loss* and *Coverage* under biased complementary labels. As we can see, ComRank shows competitive results among all methods and outperforms most datasets.

Table 8: Experimental results (mean±std) on the training data with uniform complementary labels. The best performance of each dataset is shown in **boldface**, where ↓ / ↑ indicates that smaller/larger values of metric are better performance.

| Methods | L-UW | CCMN | PMLMD | PARD | MAE | GDF | $R_u(\boldsymbol{g})$ | ComRank |
|---|---|---|---|---|---|---|---|---|
| | | | | *Coverage↓* | | | | |
| scene | .437±.022 | .404±.021 | .388±.033 | .189±.020 | .412±.020 | .157±.016 | .172±.021 | **.143±.013** |
| yeast | .565±.016 | .664±.068 | .509±.033 | .622±.066 | .548±.017 | .534±.015 | .569±.021 | **.520±.019** |
| enron | .633±.059 | .652±.030 | .477±.106 | .602±.052 | .589±.060 | .470±.049 | .602±.060 | **.458±.051** |
| rcv1-s1 | .343±.027 | .472±.042 | .537±.034 | .414±.087 | .381±.036 | .297±.028 | .395±.014 | **.305±.060** |
| bibtex | .499±.014 | .463±.033 | .424±.104 | .299±.013 | .438±.016 | .218±.017 | .322±.018 | **.195±.015** |
| bookmark | .289±.007 | .395±.022 | .669±.028 | .275±.007 | .299±.007 | .210±.005 | .294±.009 | **.199±.005** |
| nuswideBoW | .425±.010 | .473±.037 | .385±.036 | .369±.011 | .409±.010 | .313±.011 | .309±.012 | **.298±.011** |
| | | | | *Ranking Loss↓* | | | | |
| scene | .523±.021 | .466±.023 | .467±.048 | .172±.033 | .488±.023 | .153±.009 | .175±.011 | **.136±.011** |
| yeast | .245±.013 | .294±.022 | .221±.018 | .355±.115 | .231±.014 | .219±.014 | .250±.015 | **.215±.013** |
| enron | .438±.028 | .481±.034 | .262±.030 | .370±.031 | .387±.034 | .231±.029 | .395±.024 | **.221±.029** |
| rcv1-s1 | .272±.032 | .344±.044 | .447±.044 | .299±.048 | .307±.030 | .275±.058 | .369±.017 | **.267±.081** |
| bibtex | .491±.009 | .479±.031 | .460±.102 | .280±.013 | .426±.009 | .206±.013 | .308±.014 | **.180±.012** |
| bookmark | .280±.007 | .368±.032 | .661±.024 | .269±.009 | .287±.006 | .196±.005 | .280±.008 | **.187±.007** |
| nuswideBoW | .287±.005 | .340±.018 | .274±.026 | .317±.094 | .274±.005 | .206±.005 | .196±.006 | **.192±.006** |
| | | | | *One Error↓* | | | | |
| scene | .851±.017 | .758±.035 | .792±.057 | .411±.021 | .803±.024 | .384±.020 | .418±.022 | **.352±.014** |
| yeast | .252±.024 | .261±.035 | .270±.038 | .324±.202 | .251±.023 | .249±.025 | .277±.017 | **.249±.025** |
| enron | .723±.060 | .781±.058 | .519±.136 | .588±.074 | .647±.075 | .330±.116 | .668±.074 | **.308±.124** |
| rcv1-s1 | .708±.024 | .708±.059 | .750±.072 | .624±.144 | .702±.043 | .652±.085 | .777±.031 | **.609±.174** |
| bibtex | .915±.010 | .894±.024 | .881±.033 | .720±.025 | .871±.015 | .480±.023 | .744±.018 | **.418±.021** |
| bookmark | .617±.011 | .745±.040 | .927±.016 | .550±.011 | .600±.011 | .483±.008 | .608±.007 | **.474±.006** |
| nuswideBoW | .643±.019 | .710±.040 | .690±.066 | .630±.044 | .629±.023 | .540±.023 | **.488±.017** | .500±.024 |

Table 9: Experimental results (mean±std) on the training data with biased complementary labels. The best performance of each dataset is shown in **boldface**, where ↓ / ↑ indicates that smaller/larger values of metric are better performance.

| Methods | L-UW | CCMN | PMLMD | PARD | MAE | GDF | $R_u(\boldsymbol{g})$ | ComRank |
|---|---|---|---|---|---|---|---|---|
| | | | | *Coverage↓* | | | | |
| scene | .450±.018 | .402±.019 | .405±.039 | .160±.028 | .422±.020 | .144±.008 | .162±.009 | **.130±.010** |
| yeast | .575±.018 | .648±.055 | **.514±.029** | .712±.081 | .557±.019 | .540±.021 | .582±.025 | .530±.019 |
| enron | .634±.059 | .681±.036 | .494±.040 | .587±.067 | .600±.062 | .467±.060 | .603±.060 | **.456±.058** |
| rcv1-s1 | .348±.049 | .425±.038 | .524±.049 | .397±.047 | .384±.042 | .354±.050 | .441±.008 | **.350±.061** |
| bibtex | .497±.012 | .485±.026 | .463±.103 | .300±.011 | .437±.012 | .230±.015 | .327±.017 | **.206±.014** |
| bookmark | .299±.007 | .381±.029 | .655±.024 | .289±.009 | .305±.007 | .219±.006 | .300±.008 | **.210±.008** |
| nuswideBoW | .404±.011 | .449±.020 | .384±.037 | .451±.115 | .391±.011 | .313±.011 | .307±.012 | **.298±.011** |
| | | | | *Ranking Loss↓* | | | | |
| scene | .506±.025 | .466±.025 | .447±.039 | .208±.022 | .476±.024 | .171±.016 | .188±.024 | **.154±.014** |
| yeast | .240±.011 | .298±.040 | .216±.026 | .277±.034 | .228±.011 | .218±.011 | .241±.011 | **.212±.011** |
| enron | .439±.029 | .452±.043 | .250±.055 | .386±.039 | .377±.034 | .233±.029 | .396±.030 | **.221±.024** |
| rcv1-s1 | .271±.018 | .383±.033 | .452±.036 | .321±.103 | .305±.027 | .223±.035 | .323±.026 | **.230±.070** |
| bibtex | .493±.012 | .455±.034 | .423±.115 | .278±.011 | .427±.013 | .194±.015 | .303±.016 | **.170±.013** |
| bookmark | .270±.007 | .386±.025 | .677±.028 | .255±.007 | .281±.007 | .186±.005 | .274±.009 | **.175±.005** |
| nuswideBoW | .309±.005 | .363±.040 | .274±.024 | .256±.010 | .293±.004 | .208±.006 | .200±.006 | **.193±.006** |
| | | | | *One Error↓* | | | | |
| scene | .852±.026 | .749±.033 | .807±.052 | .482±.031 | .820±.018 | .435±.030 | .447±.036 | **.402±.024** |
| yeast | .254±.024 | .254±.029 | .267±.034 | .293±.127 | .252±.025 | **.253±.024** | .265±.025 | .255±.025 |
| enron | .727±.058 | .764±.073 | .493±.108 | .632±.104 | .627±.079 | .341±.136 | .669±.068 | **.324±.144** |
| rcv1-s1 | .693±.032 | .704±.080 | .737±.085 | .568±.125 | .716±.053 | .611±.038 | .724±.068 | **.561±.150** |
| bibtex | .917±.011 | .886±.033 | .811±.126 | .717±.024 | .873±.014 | .461±.025 | .747±.019 | **.414±.029** |
| bookmark | .589±.016 | .755±.040 | .926±.013 | .546±.011 | .582±.011 | .469±.008 | .604±.016 | **.448±.009** |
| nuswideBoW | .649±.018 | .719±.058 | .668±.055 | .608±.026 | .636±.021 | .537±.027 | **.490±.017** | .496±.023 |

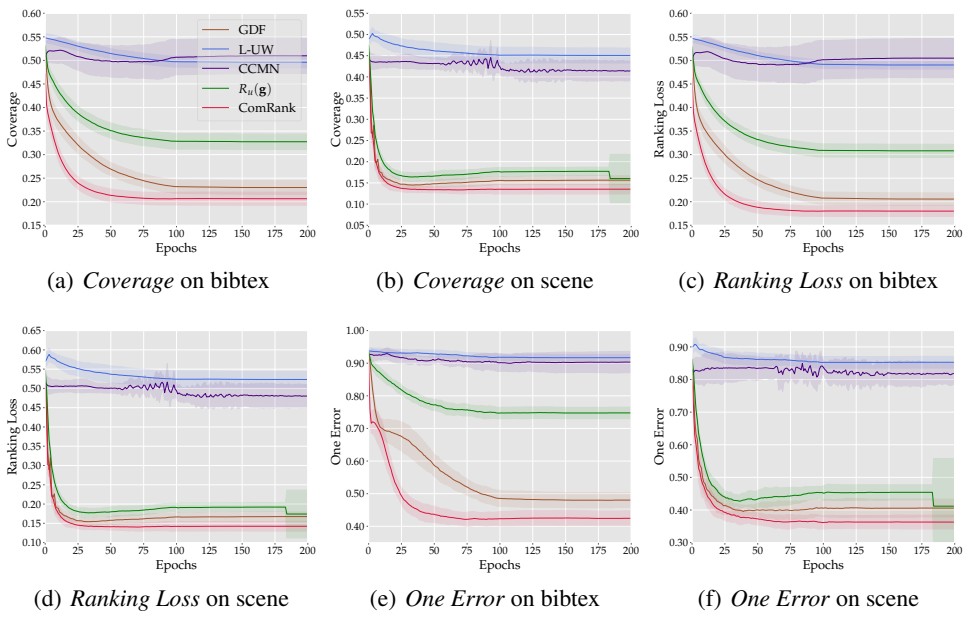

(a) *Coverage* on bibtex     (b) *Coverage* on scene     (c) *Ranking Loss* on bibtex

(d) *Ranking Loss* on scene     (e) *One Error* on bibtex     (f) *One Error* on scene

Figure 2: *Coverage*, *Ranking Loss* and *One Error* on various datasets with uniform complementary labels.

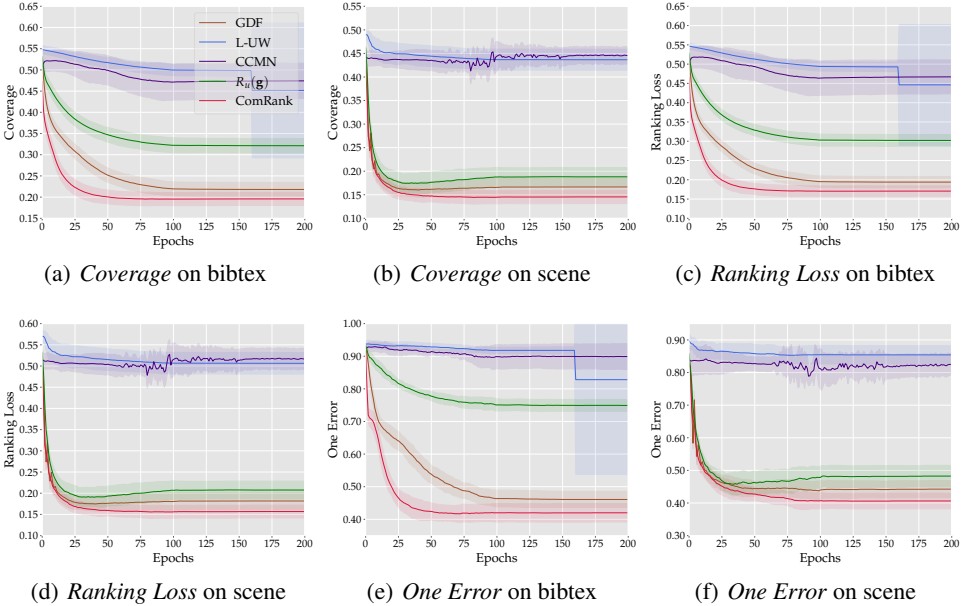

(a) *Coverage* on bibtex     (b) *Coverage* on scene     (c) *Ranking Loss* on bibtex

(d) *Ranking Loss* on scene     (e) *One Error* on bibtex     (f) *One Error* on scene

Figure 3: *Coverage*, *Ranking Loss* and *One Error* on various datasets with biased complementary labels. Dark lines show the mean of testing results, where light shadows correspond to the std.

## G Figure

Figure 2 and Figure 3 illustrate the epoch-wise trends of *Coverage*, *Ranking Loss* and *One Error* for CCMN, L-UW, GDF, $R_u(\mathbf{g})$, and ComRank, for both uniform complementary labels and biased complementary labels. As observed, ComRank consistently outperforms other methods, displaying the most rapid and least pronounced decline in *Coverage*, *One Error* and *Ranking Loss*. Notably, GDF, $R_u(\boldsymbol{g})$, and CCMN exhibit slight instability, with fluctuations emerging either at the initial stages or towards the end of the descent. In contrast, ComRank maintains remarkable stability, underscoring its effectiveness in the gradient descent process whenever complementary labels are selected uniformly or biasedly.

# H    Surrogate Loss Ablation.

The Table 10 reports the *Average Precision* from an ablation study on different surrogate losses under uniform complementary labels. The surrogate losses include:

**Log loss**: $\bar{\mathcal{L}}(\boldsymbol{g}(\boldsymbol{x}), \bar{y}) = \sum_{k=1, k \neq \bar{y}}^{K} log \left(1 + (\boldsymbol{g}_{\bar{y}}(\boldsymbol{x}) - \boldsymbol{g}_k(\boldsymbol{x}))\right)$.

**Sigmoid loss**: $\bar{\mathcal{L}}(\boldsymbol{g}(\boldsymbol{x}), \bar{y}) = \sum_{k=1, k \neq \bar{y}}^{K} (\boldsymbol{g}_{\bar{y}}(\boldsymbol{x}) - \boldsymbol{g}_k(\boldsymbol{x}))$. Since the predicted probability $\boldsymbol{g}$ in MLL is already produced through an output sigmoid function to keep $\boldsymbol{g}$ in [0,1], we directly use the difference between $\boldsymbol{g}$ values.

**Softmax loss**: $\bar{\mathcal{L}}(\boldsymbol{g}(\boldsymbol{x}), \bar{y}) = \sum_{k=1, k \neq \bar{y}}^{K} \left(\boldsymbol{g}_{\bar{y}}^{softmax}(\boldsymbol{x}) - \boldsymbol{g}_k^{softmax}(\boldsymbol{x})\right)$. Here, $\boldsymbol{g}^{softmax}$ refers to the version of $\boldsymbol{g}$ where the output function is changed from sigmoid to softmax.

Our method achieves competitive performance in most cases, validating its effectiveness and Bayes consistency.

Table 10: *Average Precision* of different surrogate losses with uniform complementary labels.

| Surrogate loss | Log | Sigmoid | Softmax | ComRank |
|---|---|---|---|---|
| scene | 0.419 | 0.687 | 0.677 | **0.785** |
| yeast | 0.712 | **0.732** | 0.720 | 0.729 |
| enron | 0.547 | 0.591 | 0.567 | **0.627** |
| rcv1-s1 | 0.263 | 0.475 | 0.280 | **0.605** |
| bibtex | 0.453 | 0.649 | 0.450 | **0.677** |
| bookmark | 0.197 | **0.629** | 0.586 | 0.618 |
| nuswideBoW | 0.485 | 0.490 | 0.488 | **0.583** |

