# OpenReview forum: "ComRank: Ranking Loss for Multi-Label Complementary Label Learning"
_NeurIPS.cc/2025/Conference — NeurIPS 2025 poster_

### Official Review · Reviewer_d4Rs · 2025-06-21

**Clarity:** 3
**Significance:** 3
**Originality:** 3
**Rating:** 4
**Confidence:** 4

**Summary:**

This work introduces a ranking loss framework for multi-label complementary label learning (MLCLL). Different from existing methods that are based on the assumption that complementary labels follow a uniform distribution, the proposed framework considers the instance-dependent assumption and introduces a complementary ranking loss for the MLCLL task.

**Questions:**

Please refer to "Strengths And Weaknesses"

**Ethical Concerns:**

["NO or VERY MINOR ethics concerns only"]

**Final Justification:**

My concerns have been addressed, and the clarifications provided helped me better understand the contribution. Therefore, I have decided to raise my overall score.

**Limitations:**

Please refer to "Strengths And Weaknesses"

**Paper Formatting Concerns:**

Please refer to "Strengths And Weaknesses"

**Quality:**

3

**Strengths And Weaknesses:**

This work introduces a ranking loss framework for multi-label complementary label learning (MLCLL). Different from existing methods that are based on the assumption that complementary labels follow a uniform distribution, the proposed framework considers the instance-dependent assumption and introduces a complementary ranking loss for the MLCLL task.

Strengths:

1.	The writing of this work is clear and easy to follow.
2.	The presented mathematical proofs are rigorous.
3.	The motivation of this work is convictive and experimental results verify the effectiveness of the proposed complementary ranking loss.

Weaknesses:

1.	My main concern focuses on the section of "Complementary Label Distribution". Regarding the uniform distribution assumption, it’s limitations illustrated in this work are easy to follow and persuasive. However, the introduced biased distribution assumption in Eq. (5) is quite challenging to follow and interpret. From my understanding from Eq. (5), $p(k={\overline y} | x)$ is inversely proportional to $p(k \in Y | x)$. However, it is quite different from the biased distribution assumption. I wonder if there is a misunderstanding on my part. It would be helpful if the authors could offer a more detailed and intuitive explanation for Eq. (5).
2.	Another concern of mine is about "Assumption 4.4". I think that Assumption 4.4 is not suitable for a multi-label learning setting, it is more suitable from the single-label learning setting. As the example provided on Page 4, Line 162, if there are more than on animals in an image, a complementary label provided by annotators cannot be solely based on the image’s features. However, Assumption 4.4 plays an important role in this work, I wonder if there is a misunderstanding on my part. It would be helpful if the authors could offer a more detailed and intuitive explanation for Assumption 4.4.

I would be willing to give a higher score if the authors can address the above-mentioned concerns in the response.

---

> ### Author Rebuttal · Authors · 2025-07-30
>
> Thank you for reviewing our paper and providing valuable insights. We appreciate your feedback and would like to respond to the weaknesses raised.
>
> **R1: My main concern focuses on the section of "Complementary Label Distribution". ... It would be helpful if the authors could offer a more detailed and intuitive explanation for Eq. (5).**
> > The core idea of Eq. (5) is to model a biased complementary label distribution. Specifically, it assumes that labels outside the relevant label set have non-uniform probabilities of being selected as complementary labels, in contrast to the uniform assumption where all irrelevant labels are equally likely to be chosen.
> >
> > Eq. (5) expresses the intuition that the probability of an irrelevant label being chosen as a complementary label depends on its correlation with the relevant labels: the more similar a label is to the relevant labels, the less likely it is to be selected as a complementary label [1]. Thus, the probability of selecting ans irrelevant label $k$ as a complementary label is modeled to be inversely proportional to its likelihood of being relevant, i.e., $p(k \in Y \mid  x)$. All the above motivates our assumption in Eq. (5):
> >
> > $$
> > p(k = \bar{y} \mid  x) \propto \frac{1}{p(k \in Y \mid  x)}, if k \notin Y
> > $$
> >
> > For example, given an image of a body of water, "desert" (a semantically distant label) is more likely to be excluded than "lake" (which is similar to a possible true label like "river"). This example illustrates how label similarity influences complementary label selection, leading to a biased (non-uniform) distribution. We will revise the manuscript to clearly explain the assumption expressed in Eq. (5), and improve the intuitive explanation.
>
> **R2: Another concern of mine is about "Assumption 4.4". ... It would be helpful if the authors could offer a more detailed and intuitive explanation for Assumption 4.4.**
>
> > Assumption 4.4 aims to reflect a more realistic and relaxed generation mechanism in MLCLL, where annotators typically do not have access to the full relevant label set when assigning a complementary label. According to previous work [2,3,4], this aligns with typical weakly supervised scenarios, where annotations rely on salient or dominant features rather than full label information.
> >
> > In some scenarios—such as an image containing multiple animals—annotators may struggle to exclude all relevant categories based solely on features. However, our assumption is motivated by the observation that annotators often eliminate obviously irrelevant labels by inspecting the input instance (e.g., excluding "building" from an image showing animals), without needing to infer the full set of relevant labels. This behavior supports modeling $\bar y$ as conditionally independent of $Y$ given $ x$.
> >
> > We adopt this assumption as a tractable approximation that reflects limited annotator knowledge and allows us to derive a theoretically sound risk estimator under biased complementary label distributions. We will revise the manuscript to more clarify this motivation.
>
> **References**
>
> [1] Gao et al., Complementary to multiple labels: A correlation-aware correction approach, TPAMI, 2024.
>
> [2] Chen et al., Beyond class-conditional assumption: A primary attempt to combat instance-dependent label noise, AAAI, 2021.
>
> [3] Xia et al., Part-dependent label noise: Towards instance-dependent label noise, NeurIPS, 2020.
>
> [4] Zhu et al., A second-order approach to learning with instance-dependent label noise, CVPR, 2021.

---

> > ### Comment · Reviewer_d4Rs · 2025-08-04
> >
> > Thank you for the authors' detailed response. My concerns have been adequately addressed, and the clarifications provided helped me better understand the contribution. Therefore, I have decided to raise my overall score.

---

> > > ### Author Response · Authors · 2025-08-04
> > > **Thank you for your response!**
> > >
> > > Thank you for your encouraging support, and thank you again for your time and many helpful comments! We will revise our paper accordingly!

---

### Official Review · Reviewer_ns2z · 2025-06-27

**Clarity:** 3
**Significance:** 3
**Originality:** 2
**Rating:** 5
**Confidence:** 5

**Summary:**

This paper proposes a novel algorithm for multi-label complementary label learning (MLCLL), supported by solid theoretical analysis and strong empirical results.

However, it suffers from certain conceptual inconsistencies (see Issue 1–2), incomplete consideration of related work (see Issue 3), and potential flaws in the experimental design (see Issues 4–5).

Overall, although I raised a lot of issues, the merits outweigh the flaws. I suggest a weak accept.

**Questions:**

See above weaknesses (especially 1–5) for necessary revisions or clarifications. The overall direction is promising, but the paper would benefit from addressing the above concerns to improve its clarity, fairness, and theoretical rigor.

**Ethical Concerns:**

["NO or VERY MINOR ethics concerns only"]

**Final Justification:**

My concerns have been resolved. The paper is technically acceptable.

**Limitations:**

yes

**Quality:**

3

**Strengths And Weaknesses:**

Strengths

1. Clear motivation. Due to practical concerns such as privacy protection, MLCLL is an important problem to study. This paper is among the first to point out that in real-world scenarios, complementary labels are not sampled uniformly at random, but tend to be semantically opposed to true labels.

2. Strong theoretical foundation. The proposed method is proven to be Bayes-consistent, which is a desirable theoretical property.

3. Extensive experiments. The paper provides a comprehensive empirical evaluation across a wide range of baselines, datasets, metrics, and ablation analyses.

Weaknesses
1. Potential contradiction in modeling assumptions. Although Assumption 4.4 (instance-dependence of complementary labels) facilitates tractable loss construction and avoids unobservable dependencies on the true label set Y, it potentially conflicts with the authors' earlier intuition that complementary labels are selected with semantic exclusion from the true labels—e.g., “desert” being chosen over “lake” when the true label is "water". This highlights a fundamental tension between model identifiability and the realism of annotator behavior modeling.

2. Theoretical comparison with URE is insufficient. While the proposed ComRank loss is theoretically proven to be Bayes-consistent in preserving the ranking order of label posterior probabilities, it does not offer an unbiased estimation of the original classification risk E[L(g(x),Y)] as URE does under the uniform assumption. Therefore, the theoretical advantage of ComRank is conditional and context-specific, and it does not universally dominate URE in terms of consistency with classification-oriented loss functions. To summarize, the paper criticizes URE for its bias but praises ComRank for its consistency. However, consistency alone does not imply superiority. Could the authors provide analysis or evidence regarding the (lack of) unbiasedness of ComRank?

3. Missing discussion on related ranking-based multi-label learning (MLL) methods.
The paper lacks a discussion of prior work on ranking-based approaches in MLL, which may lead to an overstatement of novelty. For example:[1] shows that in probabilistic MLL (PML), ranking between positive and negative labels can help disambiguate false positives. [2] shows that in MLML settings, missing labels can be assumed to rank between positive and negative labels. I suggest the authors add a paragraph in the related work section discussing ranking-based MLL methods.

Additionally, Section 3 (Preliminaries) is somewhat redundant. It uses two equations to explain what is essentially standard background. I recommend simplifying this section by presenting only one general MLL risk minimization formula and using a comparative table to illustrate the differences between MLL and MLCLL in terms of labels, learners, and loss definitions—along with symbol definitions.

4. Unrealistic experimental assumption about complementary labels.
The experiments assume that each instance has exactly one complementary label. This may not reflect realistic scenarios and could bias the evaluation in favor of the proposed method.
Specifically, the proposed method relies on pairwise ranking between positive and complementary labels. With only one complementary label, the computational cost is O(K−1), but if the number of complementary labels grows to K/2, the complexity increases to O(K^2/4). Please clarify the implications of this.

5. Evaluation metrics are incomplete.
The experiments rely solely on ranking-based metrics, while classification-based metrics are not reported. Some baselines are designed for MLL or PML, which may not be directly comparable with the proposed method. If including these baselines is necessary, more recent and ranking-aware baselines should be included to ensure fairness and meaningful comparison.

6. Minor issues in Appendix A. The #Domain refers to data source rather than the data type, all datasets used are tabular data. The #Domain of BibTeX should be Text rather than Images.

[1] Partial multi-label learning, AAAI, Vol. 32, 2018.

[2] A ranking-based problem transformation method for weakly supervised multi-label learning, PR, 2024

---

> ### Author Rebuttal · Authors · 2025-07-30
>
> Thank you for reviewing our paper and providing valuable insights. We will reorganize the preliminaries section and address the issues in Appendix A. Below, we respond to the remaining concerns raised in the review.
>
> **W1: Potential contradiction in modeling assumptions.**
>
> > We are glad to have the opportunity to clarify that Assumption 4.4 does not contradict our earlier intuition regarding semantic exclusion. In fact, both true and complementary labels are selected by annotators based on the same source of information, the observed instance $ x$. Annotators assign true labels by identifying what is clearly present in the instance, and assign complementary labels by excluding what seems clearly absent. Since both decisions depend on the same features of $ x$, an implicit semantic exclusion naturally arises between complementary and true labels, even though the annotator may not know the full true label set $Y$.
> >
> > For example, if an image contains water, an annotator may not know whether the true label is 'river' or 'lake', but they can still confidently exclude 'desert' as a complementary label based on visual cues. This illustrates that semantic exclusion behavior can emerge indirectly due to shared dependence on instance features—precisely what Assumption 4.4 formalizes. This assumption provides a practical foundation for modeling complementary label selection without requiring access to $Y$.
>
> **W2: Theoretical comparison with URE is insufficient. ... Could the authors provide analysis or evidence regarding the (lack of) unbiasedness of ComRank?**
>
> > We would like to clarify that ComRank is not intended to derive URE, nor to serve as a risk estimator. Therefore, it is not applicable to analyze ComRank in the same manner as URE-based methods, i.e., deriving unbiasedness of ComRank. In fact, ComRank and URE-based methods reflect different learning paradigms. While URE-based methods follow a risk-recovery framework that reconstructs classification risk via surrogate losses and distributional assumptions, ComRank directly optimizes a pairwise ranking loss induced by complementary labels, without relying on such assumptions. Additionally, our goal is not to claim that ComRank universally dominates URE, but to highlight a limitation of URE: it may suffer from inconsistency when complementary labels are biased. In contrast, ComRank remains Bayes consistent without requiring strong assumptions on the label distribution. We will include this discussion in the revised version.
>
> **W3: Missing discussion on related ranking-based multi-label learning (MLL) methods.**
>
> > Thank you for the insightful comment. We will add the related work of ranking-based MLL methods in the revised version. Specifically, both [1,2] you suggested show how ranking can serve as an effective inductive bias under weak supervision. Beyond these, ranking losses have also been explored in standard supervised MLL settings. For example, [3] and [4] studied ranking-based surrogate losses with theoretical consistency guarantees. [5] applied calibrated label ranking for multi-label prediction, and [6] investigated pairwise ranking formulations for label dependencies. We will revise the related work section to include the above discussions.
>
> **W4: The experiments assume that each instance has exactly one complementary label. This may not reflect realistic scenarios and could bias the evaluation in favor of the proposed method. If the number of complementary labels grows to K/2, the complexity increases to O(K^2/4).**
>
> > In response to this concern, we conducted additional experiments where each training instance is uniformly assigned $K/2$ complementary labels. Indeed, the computational complexity increases with the number of complementary labels. However, our implementation can avoid explicit pairwise loops to reduce the impact of the $O(K^2/4)$ complexity. The loss can be computed using matrix operations and masking, which makes it efficient even for large label spaces. To empirically validate its efficiency, we provide the **running time**(in 10² seconds) across datasets for large label spaces. As shown in the table below, ComRank achieves comparable or superior speed to most baselines, indicating strong scalability.
> >
> > | Dataset       | #Label Classes | CCMN  | L-UW  | GDF   | MAE   | $R_u(g)$ | ComRank |
> > | ------------- | -------------- | ----- | ----- | ----- | ----- | ---------- | ------- |
> > | enron         | 53             | 2.94  | 3.22  | 3.79  | 3.72  | 3.14       | 3.15    |
> > | rcv1-s1       | 101            | 4.13  | 3.67  | 4.12  | 3.56  | 3.76       | 3.64    |
> > | bibtex        | 159            | 5.11  | 3.76  | 4.07  | 3.69  | 3.89       | 3.91    |
> > | bookmark      | 208            | 33.25 | 17.11 | 17.18 | 18.68 | 17.23      | 17.19   |
> > | **avg** | -              | 11.36 | 6.94  | 7.29  | 7.41  | 7.01       | 6.97    |          |
> >
> > Additionally, the following table reports the average precision of methods under $K/2$ complementary labels, where ComRank achieves better performance than most baselines. While the complexity increases with more complementary labels, the performance gains we observe make this trade-off both acceptable and worthwhile. We will update the manuscript to include these analyses.
> >
> > | Method   | L-UW  | CCMN  | GDF   | MAE   | $R_u(g)$ | ComRank         |
> > | -------- | ----- | ----- | ----- | ----- | ----- | --------------- |
> > | scene    | 0.481 | 0.550 | 0.845   | 0.573 | 0.849 | **0.849** |
> > | yeast    | 0.715 | 0.691 | **0.751**   | 0.728 | 0.750 | 0.749|
> > | enron    | 0.481 | 0.491 | 0.666   | 0.625 |0.652 | **0.673** |
> > | rcv1-s1  | 0.450 | 0.443 | 0.655   | 0.606 | 0.687 | **0.690** |
> > | bibtex   | 0.455 | 0.504 | 0.795   | 0.693 | 0.783 | **0.799** |
> > | bookmark | 0.622 | 0.524 | 0.709   | 0.710 | 0.714 | **0.731** |
>
>  **W5.1: Classification-based metrics are not reported.**
>
> > We add hamming loss as a classification-based evaluation metric for experiments, where each instance is associated with a single complementary label that is uniformly sampled. To ensure fairness, we selected the optimal 0-1 threshold from prediction scores to prediction labels. As shown in the table below, our method achieves the best performance across all datasets, demonstrating that it also performs well under classification-oriented metrics.
> >
> > | Dataset    | CCMN  | L-UW  | MAE   | GDF   | $R_u(g)$ | ComRank         |
> > | ---------- | ----- | ----- | ----- | ----- | ---------- | --------------- |
> > | scene      | 0.242 | 0.820 | 0.820 | 0.152 | 0.163      | **0.136** |
> > | yeast      | 0.315 | 0.685 | 0.685 | 0.580 | 0.304      | **0.227** |
> > | enron      | 0.546 | 0.210 | 0.556 | 0.340 | 0.198      | **0.165** |
> > | rcv1-s1    | 0.456 | 0.812 | 0.885 | 0.420 | 0.115      | **0.111** |
> > | bibtex     | 0.596 | 0.191 | 0.870 | 0.774 | 0.080      | **0.059** |
> > | bookmark   | 0.150 | 0.770 | 0.725 | 0.916 | 0.084      | **0.063** |
> > | nuswideBoW | 0.135 | 0.846 | 0.864 | 0.865 | 0.135      | **0.124** |
>
> **W5.2: Some baselines are designed for MLL or PML, which may not be directly comparable with the proposed method. If including these baselines is necessary, more recent and ranking-aware baselines should be included to ensure fairness and meaningful comparison.**
> > We selected the following recent MLCLL and ranking-aware methods for comparison. Their details are summarized below:
> >
> > - MLCL [7]: A recent method for multi-label complementary label learning.
> > - LogRank [3]: A log-based ranking loss inspired by surrogate risk formulations.
> > - SoftmaxRank: A baseline we design by applying softmax function over predicted label scores and comparing their pairwise differences.
> >
> > We conduct experiments on datasets with uniform complementary labels, where each instance is associated with a single complementary label, and the experimental settings follow those described in the paper. As shown by the average precision results below, our method, ComRank, achieves the best performance on most datasets, further validating its proven Bayes consistency. We will include this discussion in the revised paper.
> >
> > | New baselines | MLCL  | LogRank | SoftmaxRank | ComRank         |
> > | ------------- | ----- | ------- | ----------- | --------------- |
> > | scene         | **0.814** | 0.419   | 0.677       | 0.785           |
> > | yeast         | **0.731** | 0.712   | 0.720       | 0.729           |
> > | enron         | 0.435 | 0.547   | 0.567       | **0.627** |
> > | rcv1-s1       | 0.547 | 0.263   | 0.280       | **0.605** |
> > | bibtex        | 0.248 | 0.453   | 0.450       | **0.677** |
> > | bookmark      | 0.557 | 0.197   | 0.586       | **0.618** |
> > | nuswideBoW    | 0.530 | 0.485   | 0.488       | **0.583** |
>
> **References**
>
> [1] Xie et al., Partial multi-label learning, AAAI, 2018.
>
> [2] Li et al., A ranking-based problem transformation method for weakly supervised multi-label learning, Pattern Recognition, 2024.
>
> [3] Dembczyński et al., Consistent multilabel ranking through univariate losses, ICML, 2012.
>
> [4] Li et al., Improving pairwise ranking for multi-label image classification, CVPR, 2017.
>
> [5] Zhang et al., Binary relevance for multi-label learning: An overview, Frontiers of Computer Science, 2018.
>
> [6] Fürnkranz et al., Multilabel classification via calibrated label ranking, Machine Learning, 2008.
>
> [7] Gao et al., Complementary to multiple labels: A correlation-aware correction approach, TPAMI, 2024.

---

> > ### Comment · Reviewer_ns2z · 2025-08-04
> > **Good answer**
> >
> > I appreciate your revision plan and look forward to seeing the improved camera-ready paper. Additionally, I suggest removing the lengthy and somewhat redundant Preliminaries section to keep the presentation concise.

---

### Official Review · Reviewer_dbnV · 2025-06-29

**Clarity:** 3
**Significance:** 3
**Originality:** 3
**Rating:** 3
**Confidence:** 5

**Summary:**

This paper proposed a ranking loss framework for Multi-label complementary label learning under both uniform and biased cases. The theoretical analysis is satisfying.

**Questions:**

Please see the weaknesses which need for more evidences and explanations.

Additionally, please answer the following questions:

1. The author should provide more empiricial analysis to prove "In MLCLL, based on the fact that complementary labels are known to be irrelevant, while non-complementary labels may include relevant ones, it is generally desirable for the predicted probabilities of complementary labels to rank lower than those of non-complementary labels."

2. I hope to see more experiment results on more challenging MLL benchmark datasets, including VOC and COCO, to strengthen the empirical evaluation.

3. Since the author used the exponential function as the surrogate loss, I expect to see more ablation studies about the impact of selecting different forms of surrogate loss, including log loss, sigmoid loss, and softmax loss.

**Ethical Concerns:**

["NO or VERY MINOR ethics concerns only"]

**Final Justification:**

This paper have provided a novel weakly supervised multi-label learning paradigm that addresses an important gap in current research, with demonstrated practical relevance to real-world applications. However, the discussion on the introduce of noise in the proposed instance-dependent setting is far from satisfactory to me. To my knowledge, the instance-dependent setting in complementary label learning is quite hard to feature, most of which will turn out to be a simple biased setting. As a result, I will keep my initial score.

**Limitations:**

The relationship between Theorem 4.5 (URE under biased distribution) and the proposed ComRank requires deeper explanation. The author should explicitly state the theoretical connections.

**Quality:**

3

**Strengths And Weaknesses:**

**Strengths:**
1. The paper presents a well-organized structure with clear logical flow and professional presentation, meeting the high standards of academic writing in this field.
2. The theoretical analysis demonstrates particular rigor, offering valuable insights into the limitations of Unbiased Risk Estimator (URE) within the MLCLL context. This contribution significantly enhances our theoretical understanding of the problem.
3. The work introduces a novel weakly supervised multi-label learning paradigm that addresses an important gap in current research, with demonstrated practical relevance to real-world applications.

**Weaknesses:**
1. The assertion that the proposed framework can directly capture label correlation information from rankings (Lines 45-46) requires stronger empirical validation. Including results or analyses that explicitly demonstrate this capability would strengthen the paper’s validity.
2. The relationship between Theorem 4.5 (URE under biased distribution) and the proposed ComRank requires deeper explanation. The author should explicitly state the theoretical connections.
3. The formulation of Assumption 4.4 (instance-dependent assumption) requires further clarification to ensure rigorous theoretical grounding.

---

> ### Author Rebuttal · Authors · 2025-07-30
>
> Thank you for your meticulous and insightful review of our paper. Below are our responses to the weaknesses and other questions.
>
> **W1: The assertion that the proposed framework can directly capture label correlation information from rankings (Lines 45-46) requires stronger empirical validation.**
>
> > To claims this assertion, we measure the ability of each method to preserve label correlations. We choose to compute the *Pearson correlation matrix*[1], which can reflect the overall level of inter-label correlation. We compute it for (1) the test label sets and (2) their prediction scores of each method, and measure their difference using Frobenius norm distance (**lower is better**), shown in the following table. All experimental settings remain the same as in the paper, training on data with uniform complementary labels. ComRank shows better correlation preservation than most baselines, indicating that the ranking-based design in ComRank still helps retain meaningful label dependencies.
> >
> > Notably, ComRank achieves the lowest Frobenius distance on the scene dataset, which aligns with its highest average precision score in Table 1 of the paper. This suggests that preserving label correlation is not only theoretically desirable but also empirically linked to better predictive performance. We will add this analysis in the revised version.
> >
> > | Dataset    | CCMN  | L-UW  | GDF   | MAE   | $R_u(g)$ | ComRank |
> > | ---------- | ----- | ----- | ----- | ----- | ---------- | ------- |
> > | scene      | 6.44  | 6.38  | 1.83  | 6.42  | 1.36       | 1.82    |
> > | yeast      | 4.45  | 4.68  | 5.79  | 5.92  | 3.70       | 6.34    |
> > | enron      | 14.92 | 12.88 | 14.33 | 14.84 | 3.33       | 3.65    |
> > | rcv1-s1    | 12.42 | 15.06 | 11.32 | 15.04 | 3.20       | 4.95    |
> > | bibtex     | 15.18 | 14.98 | 13.69 | 15.22 | 1.80       | 2.71    |
> > | bookmark   | 13.38 | 15.09 | 11.38 | 15.09 | 1.27       | 11.26   |
> > | nuswideBoW | 14.52 | 14.45 | 9.22  | 14.58 | 4.93       | 8.96    |
>
> **W2&L: The relationship between Theorem 4.5 (URE under biased distribution) and the proposed ComRank requires deeper explanation.**
>
> > Theorem 4.5 serves as the motivation for designing ComRank. It reveals that the URE under a biased complementary label distribution is inherently different from the URE derived under the uniform distribution assumption. Since all existing URE-based methods rely on the uniform complementary label assumption, their theoretical guarantee of unbiasedness breaks down when the label distribution is biased. This significantly limits the generality and robustness of URE-based approaches in real-world applications, where biased label distributions are common.
> >
> > To avoid strong dependency on the complementary label distribution, we propose ComRank inspired by ranking-based learning. ComRank directly encourages the model to assign lower scores to complementary labels. As shown in Theorem 5.2, this approach achieves Bayes consistency under both uniform and biased distributions. We will include this analysis in the revised version.
>
> **W3: Assumption 4.4 (instance-dependent assumption) requires further clarification to ensure rigorous theoretical grounding.**
>
> > Assumption 4.4 aims to reflect the practical setting where annotators lack access to the relevant label set $Y$ when selecting complementary labels, and base their decisions solely on the observed instance $x$. This aligns with typical weakly supervised scenarios, where annotations rely on salient or dominant features rather than full label information[2,3,4]. While instances like images containing multiple objects may challenge annotators' ability to exclude all relevant labels, empirical evidence suggests that they can reliably eliminate clearly irrelevant labels (e.g., excluding “building” from an image of animals) by inspecting the instance alone. This supports modeling $\bar{y}$ as conditionally independent of $Y$ given $x$.
>
> **Q1: Need more empiricial analysis to prove "In MLCLL, based on the fact that complementary labels are known to be irrelevant, while non-complementary labels may include relevant ones, it is generally desirable for the predicted probabilities of complementary labels to rank lower than those of non-complementary labels."**
>
> > By encouraging complementary labels to rank lower than non-complementary labels, the model learns useful information, which eventually can translate into encouraging irrelevant labels to rank lower than the relevant labels. Additionally, to support our claim, we reported the metric of *ranking loss* [5] for all methods in test sets which measures **the proportion of relevant labels that are ranked lower than irrelevant labels** (smaller values indicate better performance). Notably, ComRank consistently achieves the lowest scores, which provides strong support for the hypothesis. Considering ComRank shows best performance in our paper, the above results confirm that such a ranking structure is not only intuitively desirable but also empirically realizable by well-trained models. We will include this analysis in the revised version to strengthen the empirical analysis.
> >
> > | Dataset    | L-UW  | CCMN  | PMLMD | PARD  | MAE   | GDF   | $R_u(g)$ | ComRank         |
> > | ---------- | ----- | ----- | ----- | ----- | ----- | ----- | ---------- | --------------- |
> > | scene      | 0.523 | 0.466 | 0.467 | 0.172 | 0.488 | 0.153 | 0.175      | **0.136** |
> > | yeast      | 0.245 | 0.294 | 0.221 | 0.355 | 0.231 | 0.219 | 0.250      | **0.215** |
> > | enron      | 0.438 | 0.481 | 0.262 | 0.370 | 0.387 | 0.231 | 0.395      | **0.221** |
> > | rcv1-s1    | 0.272 | 0.344 | 0.447 | 0.299 | 0.307 | 0.275 | 0.369      | **0.267** |
> > | bibtex     | 0.491 | 0.479 | 0.460 | 0.280 | 0.426 | 0.206 | 0.308      | **0.180** |
> > | bookmark   | 0.280 | 0.368 | 0.661 | 0.269 | 0.287 | 0.196 | 0.280      | **0.187** |
> > | nuswideBoW | 0.287 | 0.340 | 0.274 | 0.317 | 0.274 | 0.206 | 0.196      | **0.192** |
>
> **Q2: More experiment results on more challenging MLL benchmark datasets, including VOC and COCO, to strengthen the empirical evaluation.**
>
> > Due to time limitations, we conducted experiments on the VOC dataset with uniform complementary labels, where the results of average precision are shown below. ResNet18 is used to classify and experimental settings are the same as in the paper. The average precision of our method surpasses all baselines, demonstrating the effectiveness of ComRank. We will update it and COCO in the revised version.
> >
> > | Method | L-UW  | CCMN  | MAE   | GDF   | $R_u(g)$ | ComRank         |
> > | ------ | ----- | ----- | ----- | ----- | ---------- | --------------- |
> > | VOC    | 0.327 | 0.390 | 0.314 | 0.432 | 0.431      | **0.436** |
>
> **Q3: Ablation studies about the impact of selecting different forms of surrogate loss, including log loss, sigmoid loss, and softmax loss.**
>
> > The following table reports the average precision from an ablation study on different surrogate losses under uniform complementary labels. The surrogate losses include:
> >
> > * **Log loss**: $\bar{\mathcal{L}}(g( x), \bar{y}) = \sum_{k=1, k \ne \bar{y}}^{K} log\left(1 + \left(g_{\bar{y}}( x) - g_k(x)\right)\right)$.
> > * **Sigmoid loss**: $\bar{\mathcal{L}}(g( x), \bar{y}) = \sum_{k=1, k \ne \bar{y}}^{K} \left(g_{\bar{y}}( x) - g_k( x)\right)$. Since predicted probability $g$ in MLL is already produced through an output sigmoid function to keep $g$ in [0,1], we directly use the difference between $g$ values.
> > * **Softmax loss**: $\bar{\mathcal{L}}(g( x), \bar{y}) = \sum_{k=1, k \ne \bar{y}}^{K} \left(g^{softmax}_{\bar{y}}( x) - g^{softmax}_k( x)\right)$. Here, $g^{softmax}$ refers to the version of $g$ where the output function is changed from sigmoid to softmax.
> >
> > Our method achieves competitive performance in most cases, validating its effectiveness and proven Bayes consistency.
> >
> > | Surrogate loss | Log   | Sigmoid         | Softmax | ComRank         |
> > | -------------- | ----- | --------------- | ------- | --------------- |
> > | scene          | 0.419 | 0.687           | 0.677   | **0.785** |
> > | yeast          | 0.712 | **0.732** | 0.720   | 0.729           |
> > | enron          | 0.547 | 0.591           | 0.567   | **0.627** |
> > | rcv1-s1        | 0.263 | 0.475           | 0.280   | **0.605** |
> > | bibtex         | 0.453 | 0.649           | 0.450   | **0.677** |
> > | bookmark       | 0.197 | **0.629** | 0.586   | 0.618           |
> > | nuswideBoW     | 0.485 | 0.490           | 0.488   | **0.583** |
>
> **References**
>
> [1] Pearson et al., Mathematical contributions to the theory of evolution. III. Regression, heredity, and panmixia, Philosophical Transactions of the Royal Society of London. Series A, 1896.
>
> [2] Xia et al., Beyond class-conditional assumption: A primary attempt to combat instance-dependent label noise, AAAI, 2021.
>
> [3] Xia et al., Part-dependent label noise: Towards instance-dependent label noise, NeurIPS, 2020.
>
> [4] Yao et al., A second-order approach to learning with instance-dependent label noise, CVPR, 2021.
>
> [5] Zhang et al., A review on multi-label learning algorithms, IEEE TKDE, 2014

---

> > ### Comment · Reviewer_dbnV · 2025-08-04
> >
> > Assumption 4.4 suggests the anntation of complementary label is only dependent on the instance not the ground-truth, which may introduce noise to the complementary label. Can you add a discussion on this issue and how can you deal with it accordingly?

---

> > > ### Author Response · Authors · 2025-08-05
> > > **Thank you for your response!**
> > >
> > > Thank you for your insightful comment. Assumption 4.4 specifies that the distribution of the complementary label $\bar y$ is conditioned only on the instance $x$, i.e., $ p(\bar{y} \mid x) = p(\bar{y} \mid x, Y)$. We acknowledge that, in isolation, this assumption may raise concerns: without explicitly referencing the ground-truth label set $Y$, $\bar{y}$ might inadvertently coincide with a true label, introducing label noise.
> > >
> > > To address this, we combine Assumption 4.4 with a structural constraint—namely in the paper, that the probability of selecting a true label as a complementary label is zero, via the commonly used uniform assumption (Assumption 4.1) [1,2] and our proposed biased distribution (Assumption 4.3). That is, $p(k = \bar{y} | x) = 0$ if $k \in Y$. These assumptions jointly ensure that $\bar{y}$ remains disjoint from $Y$, thus preventing noise under the defined setting.
> > >
> > > This noise-free formulation is standard in the foundational literature of complementary label learning [1,2,3,4] and enables tractable theoretical analysis. Our goal here is to understand the idealized theoretical behavior of MLCLL methods before tackling annotation noise.
> > >
> > > That said, we fully agree that in practical annotation scenarios, such assumptions may not hold—e.g., annotators might mistakenly choose a relevant label as a complementary one due to ambiguity or fatigue. This is an important limitation and a promising direction for future research, especially toward building robust MLCLL methods under noisy or imperfect labeling conditions. Possible approaches include explicitly modeling the noise transition process, designing loss functions tolerant to mislabeled complementary annotations, or using a small set of clean labels for calibration. Such extensions may be beyond the scope of this paper. In this work, we intentionally focus on the limitations of existing methods under biased complementary label distributions and propose a new ranking-based loss that is Bayes consistent, which we view as a necessary foundation before tackling more complex, noise-prone scenarios.
> > >
> > > [1] Learning from complementary labels. NeurIPS, 2017.
> > >
> > > [2] Unbiased risk estimator to multi-labeled complementary label learning. IJCAI, 2023.
> > >
> > > [3] Discriminative complementary-label learning with weighted loss. ICML, 2021
> > >
> > > [4] Learning with multiple complementary labels. ICML, 2020.

---

> > > > ### Comment · Reviewer_dbnV · 2025-08-06
> > > >
> > > > Thanks for the response. However, all the papers, i.e., [1]-[4], are based on the instance-independent setting. In such case, it is easy to perform a noise-free assumption. Conversely, in your instance-dependent assumption, I cannot find the rationale for the noise-free assumption, i.e., Assumption 4.1 and 4.3, under the scenario that the ground-truth label is unknown to the annotator. In my opinion, the complementary setting partially stems from the privacy protection application, where the ground-truth is known but is not suitable to conceal in the training dataset. I highly encourage the authors to investigate the noisy case and consider the anti-noise techniques in the future research.

---

> > > > > ### Author Response · Authors · 2025-08-07
> > > > >
> > > > > Thank you very much for your thoughtful suggestion. In future work, we will investigate how noise-robust techniques can be incorporated into the complementary label learning framework to improve its resilience and generalizability on noisy data. We sincerely thank you for bringing up this valuable perspective, which will help guide our further research.

---

### Official Review · Reviewer_eDzB · 2025-07-01

**Clarity:** 2
**Significance:** 3
**Originality:** 3
**Rating:** 4
**Confidence:** 3

**Summary:**

The paper addresses multi-label complementary label learning. It identifies limitations in existing methods relying on unbiased risk estimators, which assume uniform distribution of complementary labels—an assumption often violated in real-world scenarios with annotation biases, and these methods also underutilize label correlations.To tackle this, the paper proposes ComRank, a ranking loss framework that encourages complementary labels to be ranked lower than non-complementary ones, capturing pairwise label relationships.
Experiments show ComRank outperforms existing methods on various datasets with both uniform and biased complementary labels. Its contributions include: analyzing why URE-based methods fail under biased distributions; introducing ranking loss to MLCLL via ComRank, which has theoretical guarantees under both uniform and biased cases and shows strong experimental performance.

**Questions:**

(1)Is there an error in defining a sample of Cartesian product used in MLL in the Preliminaries of Part 3? According to the definition in the text, it is a single label sample. Is it more accurate to define Y={1,2,..., K} as Y ∈ {0,1} ^ K .
(2)The paper claims ComRank “directly captures label correlation information from rankings”, but this mechanism is not explicitly validated. Maybe you clarify it by Comparing ComRank’s performance on datasets with strong vs. weak label correlations. If correlation strength correlates with ComRank’s improvement over baselines, it would validate the claim.

**Ethical Concerns:**

["NO or VERY MINOR ethics concerns only"]

**Final Justification:**

Given the original quality of the paper and the response experiments, this paper could be accepted.

**Limitations:**

The paper only mentions that ComRank is restricted to the single complementary label setting, with extending to multiple complementary labels noted as future work. It fails to discuss other critical limitations, such as the sensitivity to the choice of surrogate loss (the paper uses an exponential function without exploring alternatives like logistic loss) and scalability issues with large label spaces, where the pairwise comparisons in the loss function could lead to high computational costs.

**Paper Formatting Concerns:**

No issues

**Quality:**

3

**Strengths And Weaknesses:**

Strengths
(1) Strong theoretical foundation: Comprehensively analyzes URE limitations and proves Bayes consistency of ComRank under both uniform and biased complementary label distributions.
(2) Advances MLCLL: Introduces ranking loss to model label correlations, filling gaps in URE-based methods.
(3) Rigorous experiments: Uses diverse datasets with uniform/biased labels, outperforming baselines across metrics like Average Precision and Ranking Loss.
Weaknesses
(1) Lacks computational complexity analysis, critical for scaling to large datasets.
(2) Neglect label sparsity effects: Multi-label data often has sparse labels, but the paper doesn't test ComRank under varying sparsity. With extreme sparsity, few non-complementary labels may undermine its ranking logic, leaving its stability in such common real-world scenarios unproven.

---

> ### Author Rebuttal · Authors · 2025-07-30
>
> Thank you for the time and effort you devoted to reviewing our paper. The error in Q1 will be modified in the revision. Below we provide our responses to other weaknesses and questions.
>
> **W1 & L2: Lacks computational complexity analysis, critical for scaling to large datasets. Fail to discuss scalability issues with large label spaces, where the pairwise comparisons in the loss function could lead to high computational costs.**
>
> > In this paper, each instance is assigned a single complementary label, resulting in a computational complexity of $O(n(K-1))$ for the proposed ComRank method, where $n$ and $K$ are the number of instances and classes, respectively. Indeed, the computational complexity increases with the number of complementary labels. However, our implementation can avoid high computational costs caused by pairwise comparisons in the loss function. The loss can be computed using matrix operations and masking, which makes it efficient for large label spaces. To empirically validate its efficiency, we provide the running time (in 10² seconds) across datasets with large label spaces, where each instance is associated with $K/2$ complementary labels. As shown in the table below, ComRank achieves comparable or superior speed to most baselines, indicating scalability.
> >
> > | Dataset       | #Label Classes | CCMN  | L-UW  | GDF   | MAE   | $R_u(g)$ | ComRank |
> > | ------------- | -------------- | ----- | ----- | ----- | ----- | ---------- | ------- |
> > | enron         | 53             | 2.94  | 3.22  | 3.79  | 3.72  | 3.14       | 3.15    |
> > | rcv1-s1       | 101            | 4.13  | 3.67  | 4.12  | 3.56  | 3.76       | 3.64    |
> > | bibtex        | 159            | 5.11  | 3.76  | 4.07  | 3.69  | 3.89       | 3.91    |
> > | bookmark      | 208            | 33.25 | 17.11 | 17.18 | 18.68 | 17.23      | 17.19   |
> > | **avg** | -              | 11.36 | 6.94  | 7.29  | 7.41  | 7.01       | 6.97    |
>
> **W2: Neglect label sparsity effects: Multi-label data often has sparse labels, but the paper doesn't test ComRank under varying sparsity. With extreme sparsity, few non-complementary labels may undermine its ranking logic, leaving its stability in such common real-world scenarios unproven.**
>
> > To verify the effect of ComRank on the data with sparse labels, we conduct experiments on datasets with varying label densities. In the table below, the '#Density' column represents the label density of each dataset, where smaller values indicate higher sparsity. We report the average precision of different methods, which is also shown in Table 1 of the paper. ComRank achieves the best performance in all cases, with densities ranging from 0.082 to 0.303, which indicates its stability under sparse labeling. The reason lies in the pairwise ranking design, which relies on relative comparisons between complementary and non-complementary labels, rather than absolute label counts. We will include this analysis in the revised version.
> >
> > | Dataset  | #Density | L-UW  | CCMN  | MAE   | GDF   | $R_u(g)$ | ComRank         |
> > | -------- | -------- | ----- | ----- | ----- | ----- | ---------- | --------------- |
> > | scene    | 0.179    | 0.395 | 0.458 | 0.432 | 0.759 | 0.734      | **0.780** |
> > | yeast    | 0.303    | 0.685 | 0.646 | 0.698 | 0.712 | 0.679      | **0.715** |
> > | enron    | 0.189    | 0.375 | 0.337 | 0.427 | 0.620 | 0.411      | **0.634** |
> > | rcv1-s1  | 0.111    | 0.445 | 0.409 | 0.427 | 0.471 | 0.363      | **0.491** |
> > | bibtex   | 0.082    | 0.237 | 0.259 | 0.287 | 0.614 | 0.413      | **0.658** |
> > | bookmark | 0.083    | 0.506 | 0.397 | 0.506 | 0.619 | 0.512      | **0.628** |
>
> **Q2: "directly captures label correlation information from rankings" is not explicitly validated. Maybe you clarify it by Comparing ComRank's performance on datasets with strong vs. weak label correlations. If correlation strength correlates with ComRank's improvement over baselines, it would validate the claim.**
>
> > We agree that comparing performance across datasets with different label correlation strengths is intuitive. However, this approach might be confounded by uncontrolled factors such as feature difficulty, sample size, and label distribution. Specifically, label correlations are inherent properties of each dataset and cannot be adjusted or isolated within the same dataset for controlled comparison. Comparing across datasets with varying correlation strengths may thus lead to unfair or misleading conclusions.
> >
> > To address this issue in a fair and interpretable way, we instead measure the ability of each method to preserve label correlations. We compute the *Pearson correlation matrix*[1], which can reflect the overall level of inter-label correlation. We compute it for (1) the test label sets and (2) their prediction scores of each method, and measure their difference using Frobenius norm distance (**lower is better**), shown in the following table. All experimental settings remain the same as the paper, i.e., training on data with uniform complementary labels. ComRank shows better correlation preservation than most baselines, indicating that the ranking-based design in ComRank still helps retain meaningful label dependencies.
> >
> > Notably, ComRank achieves the lowest Frobenius distance on the scene dataset, which aligns with its highest average precision score in Table 1 of the paper. This suggests that preserving label correlation is not only theoretically desirable but also empirically linked to better predictive performance. We will add this analysis in the revised version.
> >
> > | Dataset    | CCMN  | L-UW  | GDF   | MAE   | $R_u(g)$ | ComRank |
> > | ---------- | ----- | ----- | ----- | ----- | ---------- | ------- |
> > | scene      | 6.44  | 6.38  | 1.83  | 6.42  | 1.36       | 1.82    |
> > | yeast      | 4.45  | 4.68  | 5.79  | 5.92  | 3.70       | 6.34    |
> > | enron      | 14.92 | 12.88 | 14.33 | 14.84 | 3.33       | 3.65    |
> > | rcv1-s1    | 12.42 | 15.06 | 11.32 | 15.04 | 3.20       | 4.95    |
> > | bibtex     | 15.18 | 14.98 | 13.69 | 15.22 | 1.80       | 2.71    |
> > | bookmark   | 13.38 | 15.09 | 11.38 | 15.09 | 1.27       | 11.26   |
> > | nuswideBoW | 14.52 | 14.45 | 9.22  | 14.58 | 4.93       | 8.96    |
>
> **L1: Fail to discuss the sensitivity to the choice of surrogate loss**
>
> > In response, we conducted experiments on different surrogate losses, including
> >
> > * **Log loss**: $\bar{\mathcal{L}}(g( x), \bar{y}) = \sum_{k=1, k \ne \bar{y}}^{K} log\left(1 + \left(g_{\bar{y}}(x) - g_k(x)\right)\right)$.
> > * **Sigmoid loss**: $\bar{\mathcal{L}}(g( x), \bar{y}) = \sum_{k=1, k \ne \bar{y}}^{K} \left(g_{\bar{y}}( x) - g_k( x)\right)$. Since predicted probability $g$ in MLL is already produced through an output sigmoid function to keep $g$ in [0,1], we directly use the difference between $g$ values.
> > * **Softmax loss**: $\bar{\mathcal{L}}(g( x), \bar{y}) = \sum_{k=1, k \ne \bar{y}}^{K} \left(g^{softmax}_{\bar{y}}( x) - g^{softmax}_k( x)\right)$. Here, $g^{softmax}$ refers to the version of $g$ where the output function is changed from sigmoid to softmax.
> >
> > The following table shows the average precision of different surrogate losses on the dataset with uniform complementary labels. Our proposed method, ComRank, achieves comparable perfomance in most cases, which also validates its effectiveness.
> >
> > | Surrogate loss | Log   | Sigmoid         | Softmax | ComRank         |
> > | -------------- | ----- | --------------- | ------- | --------------- |
> > | scene          | 0.419 | 0.687           | 0.677   | **0.785** |
> > | yeast          | 0.712 | **0.732** | 0.720   | 0.729           |
> > | enron          | 0.547 | 0.591           | 0.567   | **0.627** |
> > | rcv1-s1        | 0.263 | 0.475           | 0.280   | **0.605** |
> > | bibtex         | 0.453 | 0.649           | 0.450   | **0.677** |
> > | bookmark       | 0.197 | **0.629** | 0.586   | 0.618           |
> > | nuswideBoW     | 0.485 | 0.490           | 0.488   | **0.583** |
>
> **Reference**
>
> [1] Pearson et al., Mathematical contributions to the theory of evolution. III. Regression, heredity, and panmixia, Philosophical Transactions of the Royal Society of London. Series A, 1896.

---

> > ### Comment · Reviewer_eDzB · 2025-08-09
> >
> > I appreciate the additional analyses and experiments provided in the rebuttal, which offer some clarifications. I look forward to seeing them further refined and integrated into the camera-ready paper.

---

### Note · Authors · 2025-08-13

Dear Reviewers and Chairs,

We sincerely thank the entire review team for their thoughtful reviews and constructive feedback.

In this work, we propose ComRank, a ranking-loss framework for multi-label complementary label learning (MLCLL) that remains Bayes-consistent under both uniform and biased complementary label distributions, which overcomes the limitation of existing unbiased risk estimator (URE)-based approaches that rely on the assumption of a uniform complementary label distribution.

We greatly appreciate the reviewers’ recognition of our contributions, including:

* Novel theoretical analysis of URE’s limitations under biased complementary labels and Bayes consistency of ComRank \[**eDzB, dbnV, ns2z**];
* Advancing MLCLL via ranking loss to capture label correlations without strong distributional assumptions \[**eDzB, ns2z, d4Rs**];
* Extensive experiments showing consistent improvements on diverse datasets \[**eDzB, d4Rs**].

Most reviewer comments focused on empirical validation, scalability, and clarification of assumptions. We have addressed them in our responses and discussions by:

* Added complexity and scalability analysis, with large-label-space runtime results \[**eDzB, ns2z**];
* Quantified correlation preservation using Pearson correlation matrix distance \[**eDzB, dbnV**];
* Conducted surrogate loss ablations (log, sigmoid, softmax) \[**eDzB, dbnV**];
* Extended to VOC dataset and multi-complementary-label settings \[**dbnV, ns2z**];

Both **dbnV** and **d4Rs** raised questions about the practicality of our instance-dependent assumption. We clarified that it models a realistic mechanism in complementary label assignment and supports our theoretical analysis of URE’s limitations. **d4Rs** found this reasonable and increased their score. We appreciate **dbnV**’s suggestion to explore noisy complementary labels; while this is beyond our current scope, we view it as a valuable future direction.

After the rebuttals:

* **eDzB** looks forward to the integration for further analysis in the camera-ready paper;
* **dbnV** encouraged future noise-robust extensions;
* **ns2z** endorsed our revision plan and look forward to seeing the improved camera-ready paper;
* **d4Rs** confirmed concerns were addressed and raised the rating.

We thank the reviewers for their supportive and insightful comments. We believe the additional discussions fully address the raised concerns and will be reflected in the final revision.

Sincerely,

The Authors

---

### Decision · Program_Chairs · 2025-09-17

**Decision:**

Accept (poster)

**Comment:**

This paper makes a solid contribution to the multi-label learning literature, particularly by highlighting theoretical deficiencies in prior work on using complementary labels and providing a new framework to address them. The theoretical analysis is well-motivated, and the paper raises important points that advance understanding in this area.

During review, some concerns were raised about the theoretical framework and the strength of certain assumptions (especially Assumption 4.4), as well as the connection between the theory and the eventual algorithm. While these concerns are valid discussion points, the authors provided strong and thoughtful responses that clarified their reasoning and addressed most of the issues. Incorporating these clarifications and discussion into the final version will further strengthen the contribution.